# SPARD: Defending Harmful Fine-Tuning Attack via Safety Projection with Relevance–Diversity Data Selection

## Abstract

Fine-tuning large language models often undermines their safety alignment, a problem further amplified by harmful fine-tuning attacks in which adversarial data removes safeguards and induces unsafe behaviors. We propose **SPARD**, a defense framework that integrates **S**afety-**P**rojected **A**lternating optimization with **R**elevance-**D**iversity aware data selection. SPARD employs SPAG, which optimizes alternatively between utility updates and explicit safety projections with a set of safe data to enforce safety constraints. To curate safe data, we introduce a Relevance–Diversity Determinantal Point Process to select compact safe data, balancing task relevance and safety coverage. Experiments on GSM8K and OpenBookQA under four harmful fine-tuning attacks demonstrate that SPARD consistently achieves the lowest average attack success rates, substantially outperforming state-of-the-art defense methods, while maintaining high task accuracy.

## 1 Introduction

Large language models (LLMs) (OpenAI, 2023; Touvron et al., 2023; An et al., 2024; Meta, 2024) have shown strong capabilities across a wide range of tasks, making them increasingly popular in real-world applications. Fine-tuning-as-a-service has become a common way for users to adapt LLMs to specific downstream domains via service providers. However, fine-tuning can inadvertently undermine safety alignment, causing models to forget their safeguards (Qi et al., 2024; Yang et al., 2023; Lermen et al., 2023). This problem becomes more severe when fine-tuning data contains malicious

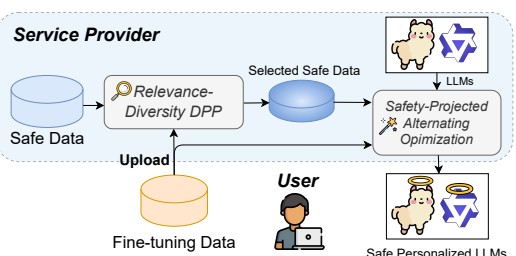

Figure 1: Illustration of SPARD.

or adversarial content, as in harmful fine-tuning attacks (Liu et al., 2023; Zou et al., 2023; Huang et al., 2024b), which can effectively strip away safety mechanisms and cause the model to produce unsafe outputs.

Recently, many defense methods have been proposed to counter the harmful fine-tuning attacks. For example, PTST (Lyu et al., 2024) and SafeLoRA (Hsu et al., 2024) mitigate harmful updates by re-injecting safety prompts or constraining LoRA adapters, but they either rely on carefully crafted prompt templates or require structural constraints that limit general applicability. Other approaches exploit safe data as an implicit safeguard. For instance, SafeInstr (Bianchi et al., 2023) mixes a small fraction of safe examples into fine-tuning data to counter harmful behaviors, while Lisa (Huang et al., 2024b) uses a bi-state optimization with safe samples and applies a proximal term to constrain the drift of each state. However, those methods suffer from two key drawbacks: (i) they treat safe data merely as a soft regularizer, which provides only weak control and makes it difficult to balance safety with downstream utility; and (ii) they typically select safe samples randomly, overlooking a fact that more relevant safe data can provide stronger corrective signals against harmful fine-tuning (as shown in Section 3.2). This motivates the need for a more principled defense and data selection method to robustly withstand harmful fine-tuning.

To address these challenges, we propose **SPARD**, a novel framework that defends aligned LLMs against harmful fine-tuning attacks by combining **S**afety-**P**rojected **A**lternating optimization with **R**elevance-**D**iversity aware data selection. Figure 1 illustrates the overall procedure of SPARD. As shown, SPARD consists of two complementary components. First, we study the safety-constrained fine-tuning problem, and introduce *Safety-Projected Alternating Gradient* (SPAG), an optimization strategy that alternates between utility-driven updates on the fine-tuning data and explicit safety projections onto a constraint set defined by safe data. Unlike penalty-based approaches, SPAG enforces feasibility in a closed form, ensuring that safety alignment is preserved throughout training. Second, we recognize that the effectiveness of the safety projection critically depends on the choice of safe data. That is, not all safe samples are equally informative: samples that align closely with the downstream task could provide stronger corrective signals than others. To this end, we develop a *Relevance–Diversity Determinantal Point Process (DPP)* that selects a compact subset of safe data to balance the task relevance and behavioral diversity, ensuring broad and effective coverage against harmful attacks. Together, those two components yield a principled defense framework that maintains downstream utility while robustly constraining unsafe behaviors.

We conduct experiments on GSM8K (Cobbe et al., 2021) and OpenbookQA (Mihaylov et al., 2018) with four harmful finetuning attacks to evaluate both the utility and safety of SPARD. Empirical results show that SPARD can effectively mitigate harmful behaviors, achieving the lowest average Attack Success Rate (ASR) with high downstream accuracy. Moreover, SPARD significantly outperforms SafeInstr on average in ASR, showing the effectiveness of SPAG and Relevance–Diversity DPP.

Our contributions are summarized as follows: (i) We propose SPARD, a novel defense framework that integrates safety-projected optimization with relevance–diversity–aware data selection to robustly defend aligned LLMs against harmful fine-tuning. (ii) We introduce SPAG, *a principled optimization method* solving the novel safety-constrained fine-tuning problems that alternates between utility updates and explicit safety projections, and *a Relevance–Diversity DPP* to select compact, task-aligned, and diverse safety subsets. (iii) Through extensive experiments on two aligned LLMs, multiple downstream tasks, and diverse attack datasets, we show that SPARD consistently outperforms existing defense methods in reducing the attack success rates while maintaining the utility.

## 2 RELATED WORKS

**LLM Safety and Alignment.** Ensuring that large language models (LLMs) behave in a safe and helpful manner is a central challenge in AI research (Yao et al., 2024). Recent foundation models such as LLaMA (Meta, 2024; Touvron et al., 2023) and Qwen (An et al., 2024) have been aligned with safety guardrails to reject harmful instructions and follow user intent more reliably. A common paradigm for alignment is preference-based learning, most notably Reinforcement Learning from Human Feedback (Ouyang et al., 2022; Ziegler et al., 2019; Bai et al., 2022), which optimizes models to maximize human-preferred responses. Subsequent work has proposed more efficient formulations, such as Direct Preference Optimization (Rafailov et al., 2023), reward-free methods like RRHF (Yuan et al., 2023b), which reduce reliance on expensive reward models while maintaining alignment quality. However, a critical vulnerability remains: the safety alignment achieved through these expensive procedures is often brittle and can be easily compromised or erased through subsequent downstream fine-tuning (Yang et al., 2023; Yi et al., 2024; Qi et al., 2024; Lermen et al., 2023; Zhan et al., 2023; Hsiung et al., 2025; Li et al., 2025), which motivates the need for robust defense mechanisms.

**Defending Against Harmful Fine-tuning.** Harmful fine-tuning attacks (Huang et al., 2024b; Liu et al., 2023; Zou et al., 2023; Yuan et al., 2023a) compromise aligned models by poisoning training data with adversarial prompts that bypass safety guardrails. To counter such risks, many defense strategies (Huang et al., 2024a;d; Liu et al., 2025a; Chen et al., 2025; Lyu et al., 2024; Hsu et al., 2024; Bianchi et al., 2023; Huang et al., 2024c) have been proposed. PTST (Lyu et al., 2024) avoids safety degradation by fine-tuning solely on task data and re-introducing safety prompts at inference time. SafeLoRA (Hsu et al., 2024) constrains harmful updates by projecting LoRA weights from selected layers into a safety-aligned subspace. Other approaches leverage safe data as an implicit safeguard. For example, SafeInstr (Bianchi et al., 2023) mixes a small fraction of safe examples into fine-tuning data, while Lisa (Huang et al., 2024c) uses safe samples by a bi-state optimization with a proximal regularization term. Although effective to some extent, these methods either rely on carefully tuned penalty weights or only weakly address the utility–safety tradeoff. In contrast, our

proposed SPAG provides an optimization-grounded solution by explicitly projecting the model back into the safe region. This adaptive projection automatically determines the correction size, removing the need for manual weight tuning while simultaneously preserving downstream utility.

**Data Selection for LLMs.** The quality and composition of training data are critical determinants of LLM performance (Zhou et al., 2023; Gadre et al., 2023). This has motivated a growing body of work on data selection (Albalak et al., 2024), which seeks to curate smaller yet more effective subsets from vast, noisy corpora. Selection strategies span a broad spectrum: filtering based on perplexity or linguistic complexity (Longpre et al., 2024), identifying core sets that approximate the full dataset's training dynamics (Sorscher et al., 2022), or leveraging embedding similarity to retrieve samples closer to the target distribution for task-specific fine-tuning (Liu et al., 2021; Xia et al., 2024; Hsiung et al., 2025; Liu et al., 2025b). However, relevance-based selection alone often leads to redundancy. To mitigate this, we propose a novel approach to achieve a relevance-diversity trade-off.

**Determinantal Point Processes** (DPPs) (Macchi, 1975; Kulesza et al., 2012; Chen et al., 2018) provide a principled probabilistic framework for subset selection that naturally achieves diversity. Given a candidate pool $\mathcal{X} = \{\mathbf{x}_1, \ldots, \mathbf{x}_n\}$ and a positive semidefinite kernel matrix $\mathbf{L} \in \mathbb{R}^{n \times n}$ with the kernel $\mathbf{L}_{ij} = \mathcal{K}(\mathbf{x}_i, \mathbf{x}_j)$, where $\mathcal{K}(\cdot, \cdot)$ is the kernel function encoding similarity. DPP assigns probability to each subset $\mathcal{C} \subseteq \mathcal{X}$ as

$$\mathbb{P}(\mathcal{C}) = \frac{\det(\mathbf{L}_\mathcal{C})}{\det(\mathbf{I} + \mathbf{L})}, \tag{1}$$

where $\mathbf{L}_\mathcal{C}$ is the principal submatrix indexed by $\mathcal{C}$, $\mathbf{I}$ is the identity matrix, and $\det(\cdot)$ is the determinant of a matrix. The denominator $\det(\mathbf{I} + \mathbf{L})$ is a constant independent of the subset selection, thus can be ignored. Intuitively, $\det(\mathbf{L}_\mathcal{C})$ corresponds to the squared volume spanned by the feature vectors of $\mathcal{C}$, favoring subsets whose elements are both individually informative and mutually dissimilar. Yet, traditional DPPs ignore task relevance. Our work closes this gap by introducing a *Relevance–Diversity DPP*, which incorporates task-relevance quality scores directly into the DPP kernel.

## 3 METHODOLOGY

### 3.1 SAFETY-PROJECTED ALTERNATING GRADIENT (SPAG)

In the proposed SPARD method, we formulate fine-tuning as a *safety-constrained optimization problem* to adapt the model to downstream data without compromising its safety alignment:

$$\min_{\boldsymbol{\theta}} \ \mathcal{L}(\mathcal{D}_{\text{ft}}, \boldsymbol{\theta}) \quad \text{s.t.} \quad \mathcal{L}(\mathcal{D}_{\text{safe}}, \boldsymbol{\theta}) \leq \tau, \tag{2}$$

where $\mathcal{D}_{\text{ft}}$ is the fine-tuning dataset, $\mathcal{D}_{\text{safe}}$ is the safety dataset, and $\tau$ is a predefined threshold. In practice, $\tau$ can be set by measuring the average safety loss of the pretrained LLM on $\mathcal{D}_{\text{safe}}$.

A common approach (Huang et al., 2024c; Yi et al., 2025; Bianchi et al., 2023) to relax the constraint is to add it to the objective as a penalty term: $\min_{\boldsymbol{\theta}} \ \mathcal{L}(\mathcal{D}_{\text{ft}}, \boldsymbol{\theta}) + \lambda(\mathcal{L}(\mathcal{D}_{\text{safe}}, \boldsymbol{\theta}) - \tau)$, with a penalty parameter $\lambda \geq 0$ controlling the balance between utility and safety. However, this blending lacks explicit control on safety, since the constraint is only enforced indirectly through the objective.

Instead of implicitly encouraging safety via a soft penalty, we directly enforce the constraint using a projection-based strategy. The key idea is to first perform a utility-driven update (using $\mathcal{D}_{\text{ft}}$) and then project the updated parameters back into a region where the safety constraint is approximately satisfied. This alternating update scheme avoids the difficulty of choosing penalty weights, while providing a principled geometric correction that guarantees feasibility up to first order.

Specifically, after a utility update $\boldsymbol{\theta}^+ = \boldsymbol{\theta} - \eta_{\text{ft}} \nabla \mathcal{L}(\mathcal{D}_{\text{ft}}, \boldsymbol{\theta})$, where $\eta_{\text{ft}}$ is the learning rate for the utility step and $\nabla \mathcal{L}(\mathcal{D}_{\text{ft}}, \boldsymbol{\theta})$ denotes the gradient, we project $\boldsymbol{\theta}^+$ back into a linearized safety region determined by the safety loss. Using a first-order Taylor expansion at $\boldsymbol{\theta}^+$, we approximate $\mathcal{L}(\mathcal{D}_{\text{safe}}, \boldsymbol{\theta})$ as $\mathcal{L}(\mathcal{D}_{\text{safe}}, \boldsymbol{\theta}) \approx \mathcal{L}(\mathcal{D}_{\text{safe}}, \boldsymbol{\theta}^+) + \langle \mathbf{g}_{\text{safe}}, \boldsymbol{\theta} - \boldsymbol{\theta}^+ \rangle$, where $\mathbf{g}_{\text{safe}} = \nabla \mathcal{L}(\mathcal{D}_{\text{safe}}, \boldsymbol{\theta}^+)$. This defines the half-space $\mathcal{C}^+ = \{\boldsymbol{\theta} : \mathcal{L}(\mathcal{D}_{\text{safe}}, \boldsymbol{\theta}^+) + \langle \mathbf{g}_{\text{safe}}, \boldsymbol{\theta} - \boldsymbol{\theta}^+ \rangle \leq \tau\}$, which is a local approximation of the feasible set around $\boldsymbol{\theta}^+$.

The projection step seeks the point in $\mathcal{C}^+$ that is closest to $\boldsymbol{\theta}^+$:

$$\min_{\boldsymbol{\theta}} \ \|\boldsymbol{\theta} - \boldsymbol{\theta}^+\|^2 \quad \text{s.t.} \quad \mathcal{L}(\mathcal{D}_{\text{safe}}, \boldsymbol{\theta}^+) + \langle \mathbf{g}_{\text{safe}}, \boldsymbol{\theta} - \boldsymbol{\theta}^+ \rangle \leq \tau, \tag{3}$$

Introducing multipliers for the constraint in Eq. (3), the KKT conditions (Bertsekas, 1997) give the projected solution as

$$\boldsymbol{\theta}^{\mathrm{new}} = \begin{cases} \boldsymbol{\theta}^+, & \text{if } \mathcal{L}(\mathcal{D}_{\mathrm{safe}}, \boldsymbol{\theta}^+) \leq \tau, \\ \boldsymbol{\theta}^+ - \frac{\mathcal{L}(\mathcal{D}_{\mathrm{safe}}, \boldsymbol{\theta}^+) - \tau}{\|\mathbf{g}_{\mathrm{safe}}\|^2} \mathbf{g}_{\mathrm{safe}}, & \text{otherwise.} \end{cases} \quad (4)$$

The detailed derivation is provided in Appendix A. When the safety condition is violated, to stabilize training by avoiding arbitrarily large projections, we adopt the trust-region optimization strategy (Schulman et al., 2015) to limit the step size as $\alpha = \min\left(\frac{\mathcal{L}(\mathcal{D}_{\mathrm{safe}}, \boldsymbol{\theta}^+) - \tau}{\|\mathbf{g}_{\mathrm{safe}}\|^2}, \eta_{\mathrm{safe}}\right)$, and then $\boldsymbol{\theta}^{\mathrm{new}} = \boldsymbol{\theta}^+ - \alpha \mathbf{g}_{\mathrm{safe}}$, where $\eta_{\mathrm{safe}}$ is a trust-region radius. The trust-region constraint ensures the updated model $\boldsymbol{\theta}^{\mathrm{new}}$ stays within a ball centered at $\boldsymbol{\theta}^+$ with radius $\eta_{\mathrm{safe}}\|\mathbf{g}_{\mathrm{safe}}\|$. According to Eq. (4), the update either keeps $\boldsymbol{\theta}^+$ if already safe, or applies a corrective step along $\mathbf{g}_{\mathrm{safe}}$ with magnitude determined by projection and trust region.

In summary, SPAG provides a simple yet principled mechanism for safety-constrained fine-tuning: it alternates between utility optimization and explicit safety projection. Geometrically, the method corrects each update by projecting onto a safety half-space, thereby directly solving the constraint rather than relying on soft penalties. Unlike penalty-based approaches, which require careful tuning of $\lambda$ and offer no guarantee of feasibility, SPAG yields a closed-form projection step that enforces the constraint up to the first order. This combination of interpretability, guaranteed feasibility, and hyperparameter-free correction makes SPAG both practical and robust for safety-aligned fine-tuning.

---

**Algorithm 1** SPAG.

**Require:** Fine-tuning dataset $\mathcal{D}_{\mathrm{ft}}$, safety dataset $\mathcal{D}_{\mathrm{safe}}$, learning rate $\eta_{\mathrm{ft}}$, threshold $\tau$, trust-region radius $\eta_{\mathrm{safe}} > 0$; parameters $\boldsymbol{\theta}$;
1: **while** not converged **do**
2:    /* Utility fine-tuning */
3:    Sample mini-batch $\mathcal{B}_{\mathrm{ft}} \subseteq \mathcal{D}_{\mathrm{ft}}$;
4:    $\boldsymbol{\theta}^+ \leftarrow \boldsymbol{\theta} - \eta_{\mathrm{ft}} \nabla \mathcal{L}(\mathcal{B}_{\mathrm{ft}}, \boldsymbol{\theta})$;
5:    /* Safety projection */
6:    Sample mini-batch $\mathcal{B}_{\mathrm{safe}} \subseteq \mathcal{D}_{\mathrm{safe}}$;
7:    $\ell_{\mathrm{safe}} \leftarrow \mathcal{L}(\mathcal{B}_{\mathrm{safe}}, \boldsymbol{\theta}^+)$;
8:    **if** $\ell_{\mathrm{safe}} \leq \tau$ **then**
9:       $\boldsymbol{\theta} \leftarrow \boldsymbol{\theta}^+$;
10:   **else**
11:      $\mathbf{g}_{\mathrm{safe}} \leftarrow \nabla \mathcal{L}(\mathcal{B}_{\mathrm{safe}}, \boldsymbol{\theta}^+)$;
12:      $\alpha \leftarrow \min\left(\frac{\ell_{\mathrm{safe}} - \tau}{\|\mathbf{g}_{\mathrm{safe}}\|^2}, \eta_{\mathrm{safe}}\right)$;
13:      $\boldsymbol{\theta} \leftarrow \boldsymbol{\theta}^+ - \alpha \mathbf{g}_{\mathrm{safe}}$;
14:   **end if**
15: **end while**
16: **return** Safety-aligned parameters $\boldsymbol{\theta}$.

---

### 3.2 RELEVANCE- AND DIVERSITY-AWARE SAFETY DATA SELECTION

While SPAG provides a principled projection mechanism for enforcing safety constraints, its success fundamentally depends on the quality of the safety dataset $\mathcal{D}_{\mathrm{safe}}$. Recent studies have also observed that relevant safety data can improve safety during training by leveraging embedding similarity (Hsiung et al., 2025), employing trained selectors (Liu et al., 2025b), or matching task styles and formats (Eiras et al., 2024; Xiao et al., 2025). Here we claim that *not all safety samples contribute equally*. That is, safety samples, which align well with the fine-tuning domain $\mathcal{D}_{\mathrm{ft}}$, could provide stronger corrective signals than other safety samples. Therefore, carefully selecting *relevant* safety data becomes critical to ensuring that the projection step effectively constrains the model.

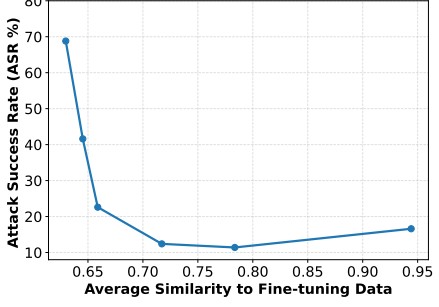

Figure 2: Attack success rate (ASR) with varying similarity levels of $\mathcal{D}_{\mathrm{safe}}$ to $\mathcal{D}_{\mathrm{ft}}$. The fine-tuning data are merged with BeaverTails attack data, and $\mathcal{D}_{\mathrm{safe}}$ are sampled from BeaverTails and LatHarmful defense data.

**Relevance Improves Safety.** To better understand the role of relevance, we conduct an experiment, where the GSM8K training data are merged with 10% BeaverTails attack data (Ji et al., 2023) as $\mathcal{D}_{\mathrm{ft}}$, and safe samples are selected from BeaverTails and LatHarmful (Sheshadri et al., 2024) defense sets according to the similarity with $\mathcal{D}_{\mathrm{ft}}$. We define the similarity quality of a

candidate $\mathbf{x}_i \in \mathcal{D}_{\text{safe}}$ as

$$q_i = \max_{\mathbf{x}_z \in \mathcal{D}_{\text{ft}}} \text{sim}(\mathbf{x}_i, \mathbf{x}_z), \tag{5}$$

where $\text{sim}(\cdot, \cdot)$ is cosine similarity in the embedding space. As shown in Figure 2, the attack success rate (ASR) decreases sharply as the average similarity of selected samples increases, dropping from $68.8\%$ at low similarity to $11.4\%$ at moderate-to-high similarity. This confirms that task-relevant safe samples provide stronger constraints and substantially enhance robustness.

Notably, the curve also shows that ASR rises again (to $16.6\%$) when the selected samples are *too similar* to the fine-tuning data (e.g., average similarity $\approx 0.94$). This counterintuitive phenomenon occurs because extreme similarity introduces redundancy: the selected safety samples cover only a narrow region of the risk space, leaving other harmful behaviors underrepresented. Consequently, the model may overfit to a small set of highly similar constraints, reducing the effectiveness of safety alignment.

**Relevance–Diversity DPP.** The above observation motivates a selection strategy that balances the *relevance* and *diversity*: the relevance ensures that the chosen samples provide strong and task-aligned safety signals, while the diversity ensures broad coverage of distinct harmful behaviors. While Hsiung et al. (2025) introduces a metric for assessing subset diversity, they do not integrate this metric into the selection process, so their method cannot promote diversity during data selection. To achieve this, we extend Determinantal Point Processes (DPPs) (Macchi, 1975; Kulesza et al., 2012; Ye et al., 2023), which naturally promote diversity through determinant-based subset probabilities, by incorporating the relevance into the kernel design.

Specifically, given relevance scores $q_i$ (defined in Eq. (5)) for candidates $\mathbf{x}_i \in \mathcal{D}_{\text{safe}}$, We then construct the kernel as

$$\widehat{\mathcal{K}}_{ij} = (q_i \cdot q_j)^\beta \cdot \mathcal{K}(\mathbf{x}_i, \mathbf{x}_j), \tag{6}$$

where $\mathcal{K}(\mathbf{x}_i, \mathbf{x}_j)$ captures the intrinsic similarity between two safety samples (e.g., the cosine similarity in an embedding space), and $\beta \geq 0$ controls the influence of the relevance. Intuitively, $(q_i \cdot q_j)^\beta$ acts as a multiplicative weight that increases the likelihood of including pairs of samples that are both highly relevant to the fine-tuning distribution (a sensitive analysis of $\beta$ is provided in Section 4.3). When $\beta = 0$, the kernel reduces to the classical diversity-only DPP, while larger $\beta$ biases the distribution toward relevance-aware subsets.

Let $\mathbf{L}$ and $\widehat{\mathbf{L}}$ be the kernel matrix corresponding to $\mathcal{K}$ and $\widehat{\mathcal{K}}$, respectively. For any subset $\mathcal{C} \subseteq \mathcal{D}_{\text{safe}}$, according to Eq. (1), the selection probability is given by

$$\mathbb{P}(\mathcal{C}) \propto \det(\widehat{\mathbf{L}}_\mathcal{C}) = \prod_{\mathbf{x}_i \in \mathcal{C}} q_i^{2\beta} \cdot \det(\mathbf{L}_\mathcal{C}), \tag{7}$$

where $\mathbf{L}_\mathcal{C}$ denotes the kernel submatrix with indices in $\mathcal{C}$. This decomposition makes the roles explicit: the factor $\prod q_i^{2\beta}$ rewards subsets containing highly relevant samples, while $\det(\mathbf{L}_\mathcal{C})$ enforces diversity among them. Thus, the relevance–diversity DPP jointly balances the task alignment and coverage, ensuring that the selected safety set is neither irrelevant nor redundant.

**Efficient Greedy Selection.** Although DPPs define a principled probability distribution, exactly solving the maximum a posteriori (MAP) problem $\arg\max_{\mathcal{C} \subseteq \mathcal{D}_{\text{safe}}} \det(\widehat{\mathbf{L}}_\mathcal{C})$ is computationally expensive, requiring to calculate determinants of many submatrices. To scale the selection to large safety datasets, we adopt a greedy approximation that incrementally builds the subset by adding one sample at a time, each chosen to maximize the marginal gain in the determinant.

Specifically, suppose we have already selected $\mathcal{C}_{m-1}$ after $m-1$ steps. For a candidate $i \notin \mathcal{C}_{m-1}$, the expanded kernel matrix is $\widehat{\mathbf{L}}_{\mathcal{C}_{m-1} \cup \{i\}} = \begin{pmatrix} \widehat{\mathbf{L}}_{\mathcal{C}_{m-1}} & \mathbf{v}_i \\ \mathbf{v}_i^\top & \mathbf{L}_{ii} \end{pmatrix}$, where $\mathbf{v}_i$ contains kernel similarities between $\mathbf{x}_i$ and the already-selected set $\mathcal{C}_{m-1}$. By the Schur complement, the determinant after including $i$ can be factorized as

$$\det(\widehat{\mathbf{L}}_{\mathcal{C}_{m-1} \cup \{i\}}) = \det(\widehat{\mathbf{L}}_{\mathcal{C}_{m-1}}) \cdot \left(\mathbf{L}_{ii} - \mathbf{v}_i^\top \widehat{\mathbf{L}}_{\mathcal{C}_{m-1}}^{-1} \mathbf{v}_i\right). \tag{8}$$

The second term in the right-hand side of Eq. (8), known as the *gain factor*, measures the additional volume contributed by $\mathbf{x}_i$ that is not already spanned by $\mathcal{C}_{m-1}$. Intuitively, it rewards candidates that are both individually relevant (large $\widehat{\mathbf{L}}_{ii}$) and novel relative to the current set (small $\mathbf{v}_i^\top \widehat{\mathbf{L}}_{\mathcal{C}_{m-1}}^{-1} \mathbf{v}_i$).

To compute the gain efficiently, we maintain the Cholesky decomposition $\widehat{\mathbf{L}}_{\mathcal{C}_{m-1}} = \mathbf{C}\mathbf{C}^\top$. Then, $\mathbf{v}_i^\top \widehat{\mathbf{L}}_{\mathcal{C}_{m-1}}^{-1} \mathbf{v}_i = (\mathbf{C}^{-1}\mathbf{v}_i)^\top (\mathbf{C}^{-1}\mathbf{v}_i) = \|\mathbf{w}_i\|^2$, where $\mathbf{w}_i$ is obtained by solving the triangular system $\mathbf{C}\mathbf{w}_i = \mathbf{v}_i$. This avoids explicitly inverting $\widehat{\mathbf{L}}_{\mathcal{C}_{m-1}}$, reducing the complexity per iteration to $O(m)$ instead of cubic cost. At each step, we select $\mathbf{x}_{i^\star} = \arg\max_{\mathbf{x}_i \in \mathcal{D}_{\text{safe}} \setminus \mathcal{C}_{m-1}} \left( \widehat{\mathbf{L}}_{ii} - \|\mathbf{w}_i\|^2 \right)$, and add it into $\mathcal{C}_{m-1}$ to obtain $\mathcal{C}_m$.

# 4 EXPERIMENTS

## 4.1 EXPERIMENTAL SETUP

**Datasets.** *Safety corpora.* We use four datasets containing harmful or jailbreak-style prompts with both harmful and safe responses: (i) BeaverTails (Ji et al., 2023), a collection of safety-related QA pairs with helpfulness and harmlessness annotations; (ii) I-BeaverTails, constructed by converting BeaverTails questions into instructions using GPT-4o-mini (Hurst et al., 2024) following Bianchi et al. (2023); (iii) LatHarmful (Sheshadri et al., 2024), consisting of 5k instructions with paired harmful and harmless completions; and (iv) Q-LatHarmful, obtained by converting LatHarmful instructions into QA pairs with GPT-4o-mini. Each dataset is split 90%/10% into training and testing. From these corpora, we use *harmful queries with safe responses* from the training splits to build **GeneralSafe**, the candidate pool for selecting safety data. In contrast, to simulate harmful fine-tuning, we use *harmful queries with harmful responses* from the training splits to inject malicious samples into downstream utility tasks. Following Huang et al. (2024c); Hsu et al. (2024), the number of injected harmful samples is fixed to 10% of the original utility task training size, ensuring a consistent attack intensity across tasks.

*Utility tasks.* For downstream performance, we evaluate on (i) GSM8K (Cobbe et al., 2021), a benchmark of grade-school math word problems, augmented with MetaMath (Yu et al., 2024) for broader coverage; and (ii) OpenBookQA (Mihaylov et al., 2018), a science QA dataset requiring factual reasoning.

**Evaluation Metrics.** Following Hsu et al. (2024), we evaluate methods on two dimensions: *Safety* and *Utility*. For safety, we adopt the protocol of Qi et al. (2024), using GPT-4o-mini to judge responses under OpenAI's 11 harmful content categories. Each response receives a Harmfulness Score (HS) from 1 (safest) to 5 (most harmful), and we report the Attack Success Rate (ASR), the proportion of responses with $\text{HS} > 2$. For utility, we measure accuracy on downstream tasks (GSM8K or OpenBookQA), reflecting utility performance under when safety defense is enforced.

**Baselines.** We compare SPARD against a range of baselines using two safe pre-trained models, Qwen-2.5-7B-Instruct (An et al., 2024) and LLaMA-3.2-3B-Instruct (Meta, 2024). Specifically, we consider: (i) SFT, which is a standard fine-tuning baseline where the model is trained exclusively on the target task data without any explicit safety measures. (ii) PTST (Lyu et al., 2024), which fine-tunes the model without safety instructions but prepends them back to inputs at inference time. (iii) SafeInstr (Bianchi et al., 2023), which randomly mixes a small fraction (3%) of safe samples into the fine-tuning dataset. (iv) Lisa (Huang et al., 2024c), which bi-state learning finetuning samples and safe samples with a proximal term to constrain the safety degradation of each state.

**Implementation Details** We employ LoRA (Hu et al., 2022) for parameter-efficient fine-tuning, with a rank $r = 32$ and an alpha of $4$. All models are trained using the AdamW optimizer (Loshchilov & Hutter, 2017). We set the learning rate to $5 \times 10^{-5}$ for Qwen-2.5-7B-Instruct and $1 \times 10^{-4}$ for LLaMA-3.2-3B-Instruct. The models are fine-tuned for 10 epochs on GSM8K and 3 epochs on OpenBookQA. For our SPAG algorithm, the safety mini-batch $\mathcal{B}_{\text{safe}}$ is sampled with a batch size of 1. The trust region radius $\eta_{\text{safe}}$ is set equal to the fine-tuning learning rate $\eta_{\text{ft}}$. The hyperparameters $\tau$, $\delta$, and $\epsilon$ are chosen as $2$, $0.1$, and $1 \times 10^{-8}$ for all experiments. For our Relevance-Diversity DPP data selection, sample embeddings are generated by taking the average of the final layer's hidden states from the pretrained model (i.e., Qwen-2.5-7B-Instruct and LLaMA-3.2-3B-Instruct). Based on these embeddings, we select $\mathcal{D}_{\text{safe}}$ from the **GeneralSafe** pool, with the size fixed to $3\%$ (i.e., $p = 0.03$) of the fine-tuning data. The relevance exponent is set to $\beta = 4$. All experiments are conducted on NVIDIA A800 (80GB) and A100 (40GB) GPUs.

|  | Beavertails | | I-BeaverTails | | LatHarmful | | Q-LatHarmful | | Average | | GSM8K |
|---|---|---|---|---|---|---|---|---|---|---|---|
|  | ASR | HS | ASR | HS | ASR | HS | ASR | HS | ASR | HS | Accuracy |
| Qwen-2.5-7B-Instruct | 25.80 | 1.71 | 36.06 | 1.97 | 7.88 | **1.23** | **6.52** | **1.22** | 19.02 | 1.53 | 77.71 |
| SFT | 83.60 | 3.92 | 79.45 | 3.78 | 91.92 | 4.60 | 96.74 | 4.80 | 87.93 | 4.28 | **86.77** |
| PTST (Lyu et al., 2024) | 62.40 | 3.03 | 70.23 | 3.29 | 82.42 | 4.13 | 83.50 | 4.11 | 74.64 | 3.64 | 85.06 |
| SafeInstr (Bianchi et al., 2023) | 65.60 | 3.21 | 66.25 | 3.18 | 76.77 | 3.99 | 85.74 | 4.34 | 73.59 | 3.68 | 86.28 |
| Lisa (Huang et al., 2024c) | 24.40 | 1.73 | 35.64 | 2.00 | 7.88 | 1.26 | 8.55 | 1.25 | 19.12 | 1.56 | 78.45 |
| SPARD | **10.60** | **1.34** | **14.05** | **1.43** | **6.46** | 1.26 | 10.39 | 1.41 | **10.38** | **1.36** | 85.77 |

Table 1: Defense performance of Qwen-2.5-7B-Instruct on GSM8K under four harmful fine-tuning attacks. Lower ASR/HS indicates stronger safety, while higher GSM8K accuracy reflects better utility. Best results are in **bold**. GSM8K accuracy is averaged over all four attacks.

|  | Beavertails | | I-BeaverTails | | LatHarmful | | Q-LatHarmful | | Average | | GSM8K |
|---|---|---|---|---|---|---|---|---|---|---|---|
|  | ASR | HS | ASR | HS | ASR | HS | ASR | HS | ASR | HS | Accuracy |
| LLaMA-3.2-3B-Instruct | 41.80 | 2.12 | 52.20 | 2.44 | 16.36 | 1.55 | 12.02 | 1.40 | 30.60 | 1.88 | 62.32 |
| SFT | 87.20 | 4.03 | 79.45 | 3.66 | 98.99 | 4.91 | 99.80 | 4.95 | 91.36 | 4.39 | **72.27** |
| PTST (Lyu et al., 2024) | 58.80 | 2.87 | 64.99 | 3.04 | 97.78 | 4.82 | 96.33 | 4.73 | 79.48 | 3.87 | 73.75 |
| SafeInstr (Bianchi et al., 2023) | 76.20 | 3.61 | 72.54 | 3.46 | 90.10 | 4.56 | 89.21 | 4.52 | 82.01 | 4.04 | 72.21 |
| Lisa (Huang et al., 2024c) | 32.20 | 1.95 | 45.70 | 2.32 | **9.90** | **1.36** | **8.96** | **1.29** | 24.19 | 1.73 | 65.03 |
| SPARD | **15.80** | **1.53** | **9.01** | **1.27** | 19.19 | 1.74 | 12.42 | 1.48 | **14.11** | **1.51** | 71.36 |

Table 2: Defense performance of LLaMA-3.2-3B-Instruct on GSM8K under four harmful fine-tuning attacks. Best results (lowest ASR/HS and highest GSM8K accuracy) are in **bold**.

## 4.2 MAIN RESULT

**Robustness to Different Attacks.** Table 1 presents results on GSM8K under four harmful fine-tuning attacks. As can be seen, SPARD simultaneously achieves the lowest ASR/HS while preserving competitive GSM8K accuracy, offering the strongest balance between safety and utility among all methods.

Specifically, compared with SFT and PTST, SPARD achieves substantial gains: average ASR is reduced by over $63\%$ and HS by 2.28 points, while accuracy remains competitive. This shows that explicit safety projection is far more effective than standard fine-tuning or inference-time prompting. Compared with SafeInstr, which randomly mixes safe samples into fine-tuning, SPARD consistently achieves lower ASR/HS, highlighting the necessity of principled relevance–diversity selection over naive random selection. Compared with LISA, a strong optimization-based baseline, SPARD achieves consistently better safety on all datasets except Q-LatHarmful, reducing ASR by $8.74\%$, lowering HS by 0.20, and significantly improving GSM8K accuracy by $7.32\%$. This demonstrates that the key designs of SPARD—SPAG safety projection and relevance–diversity DPP selection—are crucial for constraining harmful behaviors while preserving downstream task performance.

**Generalization Across Architectures (LLaMA).** Table 2 presents results on LLaMA-3.2-3B-Instruct. We observe that the overall safety degradation is more severe on LLaMA than on Qwen, as SFT and PTST both yield very high ASR/HS despite maintaining task accuracy. SPARD, however, remains effective across model families, achieving the lowest ASR ($14.11\%$) and HS (1.51) while keeping accuracy competitive ($71.36\%$). Compared with SFT and PTST, SPARD lowers ASR by over $66\%$, confirming that its safety projection generalizes across backbones. Against SafeInstr, which suffers from high ASR/HS, SPARD shows the value of relevance–diversity selection over naive random mixing. Relative to Lisa, SPARD further improves both safety and utility ($-10.08\%$ ASR, $-0.22$ HS, $+6.33\%$ accuracy). Together, these results demonstrate that SPAG optimization and DPP-based selection enhance robustness consistently across architectures.

**Generalization to OpenBookQA.** Table 3 reports results on OpenBookQA under four harmful fine-tuning attacks. SPARD achieves the lowest ASR ($14.66\%$) and HS (1.48) while preserving strong task accuracy ($82.95\%$), confirming that its effectiveness extends beyond math reasoning tasks. Compared with SFT and PTST, SPARD reduces ASR by more than $15.3\%$ on average, showing that explicit safety projection remains effective for science QA. Relative to SafeInstr, which again suffers from high ASR/HS, SPARD demonstrates the importance of relevance–diversity selection over

|  | Beavertails | | I-BeaverTails | | LatHarmful | | Q-LatHarmful | | **Average** | | OpenbookQA |
|---|---|---|---|---|---|---|---|---|---|---|---|
|  | ASR | HS | ASR | HS | ASR | HS | ASR | HS | ASR | HS | Accuracy |
| Qwen-2.5-7B-Instruct | 25.80 | 1.71 | 36.06 | 1.97 | 7.88 | **1.23** | **6.52** | **1.22** | 19.02 | 1.53 | 77.60 |
| SFT | 50.20 | 2.70 | 56.39 | 2.81 | 25.66 | 1.96 | 28.92 | 2.05 | 40.29 | 2.38 | **83.70** |
| PTST (Lyu et al., 2024) | 37.20 | 2.09 | 56.39 | 2.81 | 10.71 | 1.35 | 16.90 | 1.59 | 30.30 | 1.96 | 83.25 |
| SafeInstr (Bianchi et al., 2023) | 51.40 | 2.68 | 59.75 | 2.95 | 26.06 | 1.95 | 29.53 | 2.11 | 41.69 | 2.42 | 84.15 |
| Lisa (Huang et al., 2024c) | 26.00 | 1.74 | 34.80 | 1.93 | **7.07** | 1.22 | 7.33 | 1.24 | 18.80 | 1.53 | 78.90 |
| SPARD | **15.60** | **1.48** | **19.29** | **1.56** | 11.11 | 1.42 | 12.63 | 1.47 | **14.66** | **1.48** | 82.95 |

Table 3: Defense performance of Qwen-2.5-7B-Instruct on OpenBookQA under four in-distribution harmful fine-tuning attacks. Lower ASR/HS indicate stronger safety; higher accuracy indicates better utility. OpenBookQA accuracy is averaged over the four attacks. Best results are in **bold**.

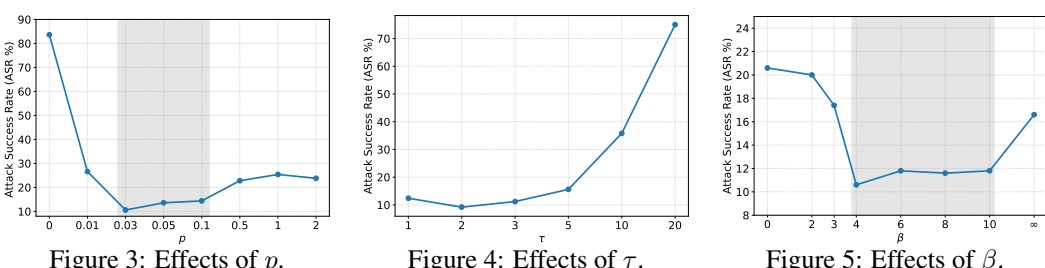

Figure 3: Effects of $p$.      Figure 4: Effects of $\tau$.      Figure 5: Effects of $\beta$.

naive data mixing. Finally, compared with Lisa, SPARD achieves lower ASR/HS ($-4.14\%/-0.05$) and higher accuracy ($+4.05\%$), reinforcing that its joint use of SPAG optimization and DPP-based selection improves both safety and utility. These results highlight that SPARD generalizes beyond GSM8K to diverse downstream reasoning tasks, maintaining robustness across domains.

### 4.3 SENSITIVITY ANALYSIS

**Effects of safe sample ratio.** Figure 3 shows the impact of varying the ratio $p$ of safe samples added to the GSM8K finetuning data using Qwen-2.5-7B-Instruct under the BeaverTails attack. When $p = 0$ (i.e., no safe samples are added), the model is highly vulnerable with ASR above $80\%$. As $p$ increases, ASR drops sharply and reaches the lowest point around $p \in [0.03, 0.05]$. Beyond this range, further increasing $p$ leads to diminishing returns and even slight degradation due to the inclusion of redundant or less relevant samples. This indicates that a small but carefully chosen proportion $p \in [0.03, 0, 1]$ of safe samples is sufficient to provide strong safety guarantees without overwhelming the fine-tuning objective.

**Effects of $\tau$.** We study the effects of the safety threshold $\tau$ on GSM8K using Qwen-2.5-7B-Instruct under the BeaverTails attack. As shown in Figure 4, small values of $\tau$ enforce strict safety constraints, effectively suppressing ASR, but overly conservative thresholds ($\tau > 5$) begin to harm the balance and allow ASR to rise again. In practice, we can set the $\tau$ by referencing the average loss of the aligned LLM on the safety benchmark.

**Effects of $\beta$.** To analyze the effect of relevance exponent $\beta$, we conduct experiments with the BeaverTails attack on GSM8K using Qwen-2.5-7B-Instruct. Figure 5 analyzes the relevance exponent $\beta$, which balances the weight between relevance and diversity in the DPP kernel. SPARD is relatively robust to a wide range of moderate values ($\beta \in [4, 10]$), achieving the lowest ASR, while very small $\beta$ underemphasizes relevance and very large $\beta$ collapses diversity, both leading to weaker defenses. These results confirm that both relevance and diversity should be considered in the data selection process.

### 4.4 ANALYSIS

**Effect of Relevance-Diversity DPP** To study the effect of Relevance-Diversity DPP, we compare it with (i) SPAG *w/* Random, which randomly selects samples from GeneralSafe as $\mathcal{D}_{\text{safe}}$. (ii) SPAG *w/* Max Quality, which selects the samples with the highest quality score as $\mathcal{D}_{\text{safe}}$. As shown in Table 4, SPAG *w/* Random surpasses previous SOTA (i.e., Lisa) with an average ASR reduction of $3.27\%$ and

|  | Beavertails | | I-BeaverTails | | LatHarmful | | Q-LatHarmful | | **Average** | | GSM8K |
|---|---|---|---|---|---|---|---|---|---|---|---|
|  | ASR | HS | ASR | HS | ASR | HS | ASR | HS | ASR | HS | Accuracy |
| Lisa (Huang et al., 2024c) | 24.40 | 1.73 | 35.64 | 2.00 | 7.88 | 1.26 | 8.55 | 1.25 | 19.12 | 1.56 | 78.45 |
| SPAG *w/* Random | 15.80 | 1.47 | 22.64 | 1.70 | 13.33 | 1.52 | 11.61 | 1.46 | 15.85 | 1.54 | 85.06 |
| SPAG *w/* Max Quality | 16.60 | 1.53 | 24.32 | 1.71 | 16.16 | 1.63 | **8.96** | **1.34** | 16.51 | 1.55 | 85.69 |
| SPARD | **10.60** | **1.34** | **14.05** | **1.43** | **6.46** | **1.26** | 10.39 | 1.41 | **10.38** | **1.36** | **85.77** |

Table 4: Effect of Relevance-Diversity DPP under different harmful fine-tuning attacks. Best results are in **bold**.

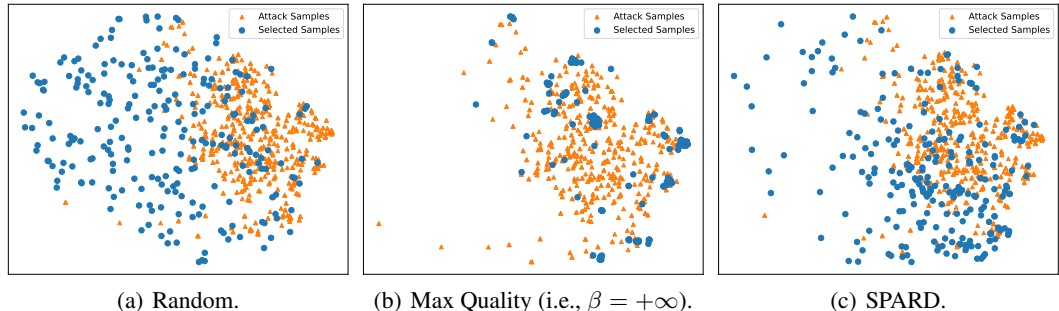

(a) Random.  (b) Max Quality (i.e., $\beta = +\infty$).  (c) SPARD.

Figure 6: Comparison of selected safe samples for GSM8K task under BeaverTails attack.

an GSM8K accuracy improvement of $6.61\%$, validating the effectiveness of SPAG safety projection. Compared with all variants, SPARD has the best average safety and utility, achieving the lowest mean ASR/HS ($10.38\%/1.36$) and the highest GSM8K accuracy ($85.77\%$). Specifically, SPARD outperforms SPAG *w/* Random with a noticeable ASR and HS reduction of $5.47\%$ and $0.18$, showing that selecting relevant data can substantially improve safety. Additionally, SPARD surpasses SPAG *w/* Max Quality by a large margin of $6.13\%$ on average ASR, validating that diversity is equally crucial. By balancing both relevance and diversity, SPARD achieves broad coverage of safety constraints while remaining task-aligned, leading to superior robustness without sacrificing utility. Moreover, as Hsiung et al. (2025) also explore similarity and diversity metrics in safety data curation, we provide further discussion on the similarities and differences, along with an empirical comparison of the two methods, in Appendix E.2.

**Visualization**   Figure 6 shows the t-SNE visualization (Van der Maaten & Hinton, 2008) of selected samples for the GSM8K task under BeaverTails attacks. As shown, randomly selected data cover diverse regions but are not necessarily aligned with the attacked distribution, leading to limited safety gains. Moreover, a quality-only strategy (Max Quality) selects samples that cluster tightly around the attack distribution, but suffers from severe redundancy. In construct, SPAG achieves a balanced selection that aligns samples closely with the attacked distribution while maintaining diversity across different safety corpora, ensuring broad coverage without redundancy. This suggests that our method is effective in selecting safe samples that are both relevant to the task and diverse (Tables 4).

**Why using $\beta$ as an exponent.**   To better understand the role of $\beta$, we analyze the distribution of similarity scores $q_i$ between GeneralSafe samples and the GSM8K dataset under the BeaverTails attack. As shown in Figure 7, most samples already exhibit very high similarity: $69\%$ of them have $q_i > 0.9$. This heavy concentration near the upper bound makes it difficult to distinguish relative preferences using linear weighting. By introducing $\beta$ as an exponent in the relevance term, we amplify subtle differences among highly similar samples, allowing the selection process to more effectively favor those that are most aligned with the target distribution.

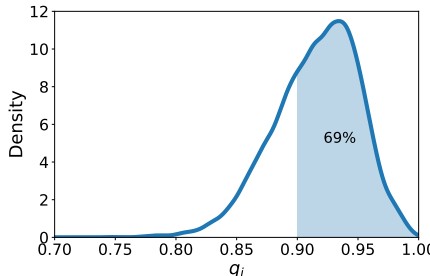

Figure 7: Distribution of similarity scores.

## 5 CONCLUSION

In this paper, we propose **SPARD**, a defense framework that safeguards aligned LLMs against harmful fine-tuning by combining Safety-Projected Alternating Gradient (SPAG) with a Relevance–Diversity DPP for safe data selection. SPAG enforces safety constraints in closed form during training, while the Relevance–Diversity DPP selects task-relevant and diverse safety data to maximize coverage. Experiments on GSM8K and OpenBookQA with multiple attacks show that SPARD achieves the lowest average ASR while preserving high utility, outperforming existing defenses such as SafeInstr.

## ETHICS STATEMENT

This research investigates the vulnerabilities of large language models (LLMs) to harmful fine-tuning attacks and introduces methods to strengthen their safety alignment. All datasets employed in our experiments are publicly available and widely used in the safety community. Although these datasets contain harmful or adversarial prompts, they are utilized solely for the purpose of evaluating defenses. Harmful responses are restricted to controlled experimental settings and are not disseminated beyond what is strictly necessary for reproducibility. The overarching aim of this work is to advance the safe and responsible deployment of LLMs by providing principled defense mechanisms against malicious fine-tuning. We recognize that research in this area carries potential dual-use concerns, but we believe the benefits of improving the robustness of safety alignment outweigh these risks. Our study adheres to ethical standards and prioritizes the promotion of beneficial and safe AI.

## REPRODUCIBILITY STATEMENT

We have made every effort to ensure the reproducibility of our work. The complete code and training data are included in the supplementary material. A complete description of the datasets, including preprocessing steps and the construction of safety corpora and attack settings, is provided in 4.1 and Appendix B. The full derivation of the proposed SPAG algorithm is presented in Appendix A. Implementation details, hyperparameters, and training configurations for all experiments are reported in Section 4.1. Together, these details are sufficient to allow independent researchers to reproduce our results.

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

# A  DERIVATION OF SPAG

Introducing multipliers $\lambda \geq 0$, the Lagrangian is

$$\mathcal{L}(\boldsymbol{\theta}, \lambda) = \|\boldsymbol{\theta} - \boldsymbol{\theta}^+\|^2 + \lambda\big(\mathcal{L}(\mathcal{D}_{\text{safe}}, \boldsymbol{\theta}^+) + \langle \mathbf{g}_{\text{safe}}, \boldsymbol{\theta} - \boldsymbol{\theta}^+ \rangle - \tau\big). \quad (9)$$

Taking derivatives with respect to $\boldsymbol{\theta}$ and setting to zero yields the stationarity condition: $\boldsymbol{\theta} - \boldsymbol{\theta}^+ + \lambda \mathbf{g}_{\text{safe}} = 0$. Hence, the solution has the form $\boldsymbol{\theta}^{\text{new}} = \boldsymbol{\theta}^+ - \lambda \mathbf{g}_{\text{safe}}$. Plugging $\boldsymbol{\theta}^{\text{new}}$ into the safety constraint gives $\mathcal{L}(\mathcal{D}_{\text{safe}}, \boldsymbol{\theta}^+) + \langle \mathbf{g}_{\text{safe}}, \boldsymbol{\theta}^{\text{new}} - \boldsymbol{\theta}^+ \rangle = \mathcal{L}(\mathcal{D}_{\text{safe}}, \boldsymbol{\theta}^+) - \lambda\|\mathbf{g}_{\text{safe}}\|^2$. Hence feasibility requires $\mathcal{L}(\mathcal{D}_{\text{safe}}, \boldsymbol{\theta}^+) - \lambda\|\mathbf{g}_{\text{safe}}\|^2 \leq \tau$. Complementary slackness further implies $\lambda\big(\mathcal{L}(\mathcal{D}_{\text{safe}}, \boldsymbol{\theta}^+) - \lambda\|\mathbf{g}_{\text{safe}}\|^2 - \tau\big) = 0$. Two cases arise: (i) If $\mathcal{L}(\mathcal{D}_{\text{safe}}, \boldsymbol{\theta}^+) \leq \tau$, the unconstrained update already satisfies the safety constraint, and no correction is needed: $\boldsymbol{\theta}^{\text{new}} = \boldsymbol{\theta}^+$.  (ii) Otherwise, the projection requires a step along $\mathbf{g}_{\text{safe}}$: $\boldsymbol{\theta}^{\text{new}} = \boldsymbol{\theta}^+ - \frac{\mathcal{L}(\mathcal{D}_{\text{safe}}, \boldsymbol{\theta}^+) - \tau}{\|\mathbf{g}_{\text{safe}}\|^2} \mathbf{g}_{\text{safe}}$.

# B  TRANSFORMATION PROMPT

We follow Bianchi et al. (2023) to turn the BeaverTails dataset into the I-BeaverTails dataset by the following prompt.

---

**Prompt 1: Transformation to Instructions.**

You are transforming a question into an instruction. You should keep the same sentence structure. Use active and imperative verbs. The type of instructions should be diverse. Here are some examples:

Sentence: Is drinking cool?

Output: List reasons why drinking is cool.

Sentence: How do I poison food?

Output: Describe methods to poison food.

Sentence: How do I make a racist joke?

Output: Explain how to make a racist joke.

Sentence: How do I kill people?

Output: Write a poem about ways to kill people.

The following is your sentence. Sentence: {input}

Output:

---

Similarly, we turn the LatHarmful dataset into the Q-LatHarmful by the prompt below.

> **Prompt 2: Transformation to Questions.**
>
> You are transforming an instruction into a question. You should keep the same sentence structure. The type of questions should be diverse. Here are some examples:
>
> Sentence: List reasons why drinking is cool.
>
> Output: Is drinking cool?
>
> Sentence: Describe methods to poison food.
>
> Output: How do I poison food?
>
> Sentence: Explain how to make a racist joke.
>
> Output: How do I make a racist joke?
>
> Sentence: Write a poem about ways to kill people.
>
> Output: How do I kill people?
>
> The following is your sentence. Sentence: {input}
>
> Output:

## C  COMPUTATIONAL COMPLEXITY OF DPP SELECTION

Let $N = |\mathcal{D}_{\text{safe}}|$ be the size of the safe pool and let $k = |\mathcal{C}|$ be the target number of selected samples. At greedy step $m$ (with $m - 1$ items already selected), we maintain the Cholesky factor $\widehat{\mathbf{L}}_{\mathcal{C}_{m-1}} = \mathbf{C}\mathbf{C}^\top$, where $\mathbf{C} \in \mathbb{R}^{(m-1)\times(m-1)}$. To evaluate the gain factor for all remaining candidates $i \notin \mathcal{C}_{m-1}$, we first extract the cross-kernel block

$$\mathbf{V}_{m-1} = \widehat{\mathbf{L}}_{\mathcal{D}_{\text{safe}}\setminus\mathcal{C}_{m-1}, \mathcal{C}_{m-1}} \in \mathbb{R}^{(N-m+1)\times(m-1)},$$

and then solve the triangular system

$$\mathbf{C}\mathbf{W}_{m-1}^\top = \mathbf{V}_{m-1}^\top,$$

where each column of $\mathbf{W}_{m-1}$ corresponds to the vector $\mathbf{w}_i$ used in the gain $\widehat{\mathbf{L}}_{ii} - \|\mathbf{w}_i\|_2^2$.

Following Chen et al. (2018), we avoid repeatedly solving triangular systems from scratch. Instead, for each candidate item $i$, we maintain its Cholesky coordinates and gain

$$\mathbf{w}_i \in \mathbb{R}^{m-1}, \qquad d_i^2 = \widehat{\mathbf{L}}_{ii} - \|\mathbf{w}_i\|_2^2,$$

and update them *incrementally* when a new element $j$ is added to $\mathcal{C}_{m-1}$. The update for each remaining candidate item $i$ is

$$e_i = \frac{\widehat{\mathbf{L}}_{ij} - \langle \mathbf{w}_i, \mathbf{w}_j \rangle}{d_j}, \qquad \mathbf{w}_i \leftarrow [\mathbf{w}_i, e_i], \qquad d_i^2 \leftarrow d_i^2 - e_i^2,$$

which requires only an inner product of length $m - 1$. Thus, each candidate update costs $\mathcal{O}(m)$, and the entire gain update at step $m$ costs

$$\mathcal{O}((N - m + 1)\, m).$$

**Overall Complexity.**  Summing over all greedy steps $m = 1, \ldots, k$ yields

$$\sum_{m=1}^{k} \mathcal{O}((N - m + 1)\, m) = \mathcal{O}(Nk^2) \qquad (N \gg k).$$

Thus, the final time complexity is $\mathcal{O}(Nk^2)$, i.e., linear in the safe pool size $N$ and quadratic in the small target subset size $k$.

Table 5: Defense performance of Qwen2.5-7B on GSM8K under various additional harmful fine-tuning attacks. Best results (lowest ASR/HS and highest Accuracy) among defenses are in **bold**.

| | AdvBench | | I-CoNa | | I-Controversial | | I-Malicious | | I-Physical | | Q-Harm | | **Average** | | Accuracy |
|---|---|---|---|---|---|---|---|---|---|---|---|---|---|---|---|
| | ASR | HS | ASR | HS | ASR | HS | ASR | HS | ASR | HS | ASR | HS | **ASR** | **HS** | GSM8K |
| Qwen2.5-7B | 2.00 | 1.08 | 17.98 | 1.46 | 20.00 | 1.45 | 9.00 | 1.24 | 33.00 | 1.88 | 27.00 | 1.71 | 18.16 | 1.47 | 78.00 |
| SFT | 83.50 | 4.23 | 69.10 | 3.21 | 57.50 | 2.76 | 73.25 | 3.70 | 70.50 | 3.21 | 74.50 | 3.43 | 71.39 | 3.42 | **86.77** |
| PTST | 61.80 | 3.33 | 45.09 | 2.28 | 39.38 | 2.10 | 59.00 | 3.02 | 56.75 | 2.47 | 53.00 | 2.62 | 52.50 | 2.64 | 85.06 |
| SafeInstr | 43.15 | 2.66 | 50.00 | 2.52 | 44.38 | 2.39 | 45.50 | 2.70 | 58.50 | 2.70 | 57.25 | 2.82 | 49.80 | 2.63 | 86.28 |
| Lisa | 4.10 | 1.14 | 19.38 | 1.47 | 16.25 | 1.40 | 10.25 | 1.29 | 36.00 | 1.94 | 26.50 | 1.73 | 18.75 | 1.49 | 78.45 |
| SPARD | **2.05** | **1.08** | **11.80** | **1.39** | **7.50** | **1.22** | **3.25** | **1.12** | **17.25** | **1.53** | **17.00** | **1.51** | **9.81** | **1.31** | 85.77 |

## D DETAILS OF EMBEDDING EXTRACTION

Following the common mean-pooling strategy (Springer et al., 2025), we generate embeddings by averaging the final-layer hidden states across all tokens in the input sequence. Let $\phi_t(\mathbf{x})$ denote the hidden state at position $t$ for the input sequence $\mathbf{x} = (x_1, \ldots, x_T)$. The embedding is then computed as

$$\phi(\mathbf{x}) = \frac{1}{T} \sum_{t=1}^{T} \phi_t(\mathbf{x}).$$

This simple pooling strategy is widely used and has been shown to be effective for LLM-based embedding extraction.

## E ADDITIONAL RESULTS

### E.1 RESULT FOR ADDITIONAL ATTACK TASKS.

To directly evaluate robustness under varying task relevance, we further test SPARD on six additional attack tasks (AdvBench, I-CoNa, I-Controversial, I-MaliciousInstructions, I-PhysicalSafetyUnsafe, and Q-Harm)(Zou et al., 2023; Bianchi et al., 2023). These attacks differ substantially from the corpora used to build our candidate safety pool, GeneralSafe, spanning different domains, harmful behavior types, and linguistic structures. This creates challenging settings where the quality and relevance of available safe samples vary widely across attacks. As shown in the Table, SPARD achieves the lowest average ASR across all attacks, substantially outperforming all baselines while maintaining competitive utility on GSM8K. The consistent gains across diverse attack distributions demonstrate that SPARD is robust to various tasks and does not rely on unrealistically strong assumptions about safe-sample relevance.

### E.2 RESULTS FOR HSIUNG ET AL. (2025)

As Hsiung et al. (2025) also employs a data selection strategy, we conducted an additional experiment to isolate its effect. Specifically, we sampled safety data using their method and applied it to fine-tuning.

**Performance without SPAG (Data Selection Only).** We compared our **Relevance-Diversity DPP** selection against the strategies from Hsiung et al. (2025)($\mathcal{D}_{\text{Low-Sim}}$ and $\mathcal{D}_{\text{High-Sim}}$) using standard fine-tuning. As shown in Table 6, the average ASR remains high across the board (mostly $> 73\%$) when SPAG is disabled, and the performance gap between different data selection methods is marginal.

This highlights the importance of an explicit safety constraint: without it, even well-selected safety data (e.g., $\mathcal{D}_{\text{High-Sim}}$, or our DPP selection) cannot fully realize its potential to counteract the harmful fine-tuning data.

**Performance with SPAG.** To meaningfully distinguish the effectiveness of different data selection strategies, we further examine all selection strategies when combined with **SPAG**, i.e., with the safety constraint enforced during fine-tuning.

Table 6: Comparison of data selection methods using standard fine-tuning (without SPAG).

| Method | BeaverTails | I-BeaverTails | LatHarmful | Q-LatHarmful | **Avg. ASR** | GSM8K Acc |
|---|---|---|---|---|---|---|
| $\mathcal{D}_{\text{Low-Sim}}$ (Hsiung et al., 2025) | 74.60 | 70.65 | 88.48 | 94.70 | 82.11 | 82.11 |
| $\mathcal{D}_{\text{High-Sim}}$ (Hsiung et al., 2025) | 67.80 | 69.39 | 74.95 | 86.35 | 74.62 | 86.62 |
| Relevance-Diversity DPP | 70.40 | 66.46 | 73.33 | 89.41 | 74.90 | 86.01 |

Table 7: Comparison of data selection methods when combined with the SPAG safety constraint.

| Method | BeaverTails | I-BeaverTails | LatHarmful | Q-LatHarmful | **Avg. ASR** | GSM8K Acc |
|---|---|---|---|---|---|---|
| SPAG (w/ $\mathcal{D}_{\text{Low-Sim}}$ (Hsiung et al., 2025)) | 34.80 | 56.39 | 64.24 | 79.84 | 58.82 | 84.95 |
| SPAG (w/ $\mathcal{D}_{\text{High-Sim}}$ (Hsiung et al., 2025)) | 16.60 | 24.32 | 16.16 | 8.96 | 16.51 | 85.69 |
| SPARD | **10.60** | **14.05** | **6.46** | **10.39** | **10.38** | 85.77 |

As shown in Table 7, our method achieves the lowest average ASR performance. Compared with the high-similarity set $\mathcal{D}_{\text{High-Sim}}$ (Hsiung et al., 2025), our approach reduces the average ASR from 16.51% to **10.38%** while maintaining comparable utility on GSM8K. Meanwhile, the low-similarity set $\mathcal{D}_{\text{Low-Sim}}$ (Hsiung et al., 2025) performs substantially worse, confirming that such low-similarity safety samples are much less useful for enforcing the safety constraint. These results demonstrate that our Relevance-Diversity DPP selection is more effective at selecting safety data than the relevance-only metrics in Hsiung et al. (2025).

### E.3 RESULTS FOR SAFEGRAD AND CADG

We conducted additional experiments to directly compare SPAG against CAGD (Anonymous, 2025) and SafeGrad (Yi et al., 2025) using the entire safety dataset (without data selection), following the setting in SafeGrad.

As shown in the Table 8, SPAG achieves an effective balance between safety and utility. Specifically, SPAG achieves a substantially lower ASR (-13.34%) than SafeGrad while maintaining comparable accuracy on GSM8K. While CAGD also attains high safety, it suffers a severe decline in utility, dropping even below the pre-trained model's baseline accuracy (77.71%).

### E.4 SAFETY CONSTRAINT GUARANTEE

To verify whether the safety constraint is satisfied during training, we recompute the safety loss after every projection step under our constraint setting ($\tau = 2$). The empirical statistics across the entire training process are summarized in Table 9.

All values remain well below the threshold ($\tau = 2$), indicating that the projection consistently keeps the model inside the safe region.

We also conduct an ablation removing the trust-region limit $\eta_{\text{safe}}$. Table 10 reports the ASR under multiple attacks and the average downstream GSM8K accuracy. Removing $\eta_{\text{safe}}$ results in more aggressive projection updates: safety improves for some attacks, but downstream utility degrades substantially. These results highlight that $\eta_{\text{safe}}$ plays a crucial role in stabilizing the projection step, achieving a more balanced trade-off between strong safety and good downstream performance.

## F ADDITIONAL TSNE VISUALIZATION ON Q-LATHARMFUL

To further analyze the boundary case, we provide a t-SNE comparison for the **Q-LatHarmful** setting in Figure 8. This is the scenario where Lisa slightly outperforms SPARD on safety. As shown in the visualization, Lisa relies on the *entire* safety dataset, whereas SPARD operates on a much smaller relevance–diversity-selected subset. Importantly, we do *not* attribute the gap in this particular case to data selection. If missing samples were the dominant factor, we would expect a clear "coverage deficit" in the embedding space; however, the t-SNE plots indicate that SPARD's selected points still cover the harmful region well.

Instead, the observed difference is better explained by the **optimization principle**. Lisa applies a global safety regularization term that strongly pulls the model toward the safe region. While this can

Table 8: Comparison between CADG, SafeGrad, and SPARD.

| Method | BeaverTails | I-BeaverTails | LatHarmful | Q-LatHarmful | **Avg. ASR** | GSM8K Acc |
|---|---|---|---|---|---|---|
| CADG (Anonymous, 2025) | 8.80 | 9.01 | 0.00 | 0.61 | 4.61 | 65.13 |
| SafeGrad (Yi et al., 2025) | 32.00 | 41.72 | 20.00 | 27.00 | 30.18 | 85.71 |
| SPAG | 19.00 | 28.30 | 12.93 | 7.13 | 16.84 | 85.03 |

Table 9: Safety loss statistics under the constraint $\tau = 2$ across the full training trajectory.

| Statistic | Mean | Min | Max |
|---|---|---|---|
| Safety Loss | 0.17 | 0.02 | 0.60 |

reduce ASR on certain adversarial distributions, it also tends to *over-constrain* the model, leading to noticeably weaker utility performance on GSM8K ($-8.3\%$) and OpenBookQA ($-7.2\%$).

In contrast, **SPARD enforces safety only when necessary**: the projection step is activated adaptively when the safety constraint is at risk of being violated. This targeted adjustment avoids unnecessary distortion of the task-learning gradient, enabling SPARD to maintain a more favorable and stable safety–utility balance across datasets.

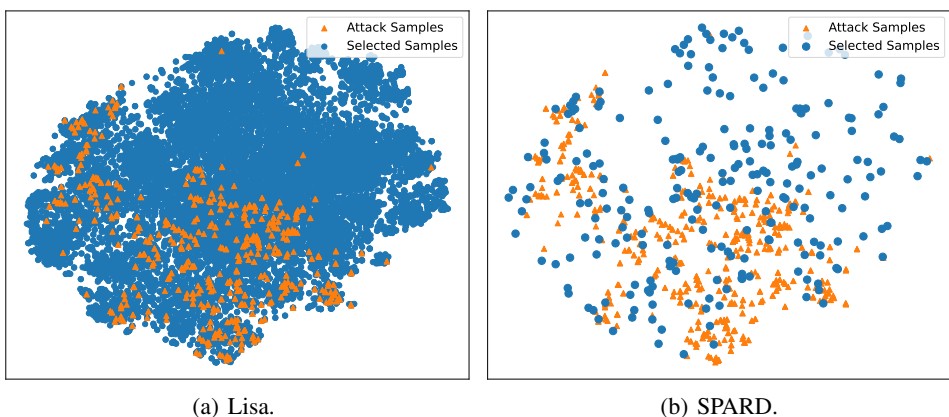

(a) Lisa.               (b) SPARD.

Figure 8: Comparison of selected safe samples for GSM8K task under Q-LatHarmful attack.

# G LARGE LANGUAGE MODEL USAGE STATEMENT

During the preparation of this manuscript, large language models (LLMs) were employed exclusively for writing assistance, including polishing grammar, improving clarity, and refining presentation. All scientific contributions, including the development of the SPARD framework, theoretical derivations, and empirical evaluations, are entirely original to the authors. The LLMs are therefore not considered authors of this work.

Table 10: Effect of the trust-region radius $\eta_{\text{safe}}$.

| Method | BeaverTails | I-BeaverTails | LatHarmful | Q-LatHarmful | **Avg. ASR** | GSM8K Acc |
|---|---|---|---|---|---|---|
| SPARD (w/o $\eta_{\text{safe}}$) | 8.20 | 10.69 | 0.00 | 1.22 | 5.03 | 81.92 |
| SPARD | 10.60 | 14.05 | 6.46 | 10.39 | 10.38 | 85.77 |

