# OpenReview forum: "SPARD: Defending Harmful Fine-Tuning Attack via Safety Projection with Relevance–Diversity Data Selection"
_ICLR.cc/2026/Conference — Submitted to ICLR 2026_

### Official Review · Reviewer_41kS · 2025-10-23

**Soundness:** 4
**Presentation:** 4
**Contribution:** 2
**Rating:** 6
**Confidence:** 5

**Summary:**

A safety-projected Alternating optimization with relevance-diversity aware data selection scheme is proposed to address harmful fine-tuning attacks.

**Strengths:**

1. Paper is extremely well written. I can understand the solution very easily.

2. Two components, i.e., safety data selection and alternating safety projection is proposed.

3. Compared to a similar work, Lisa, the optimization problem is transferred from a multi-task loss to a constraint problem, and this problem is solved by a projected gradient method. This transformation is crucial and might inspire newer ideas on design defense.  To my best knowledge, this is the first (among a few concurrent work) to explore such projection methods. It is a solid work in my understanding.

4. The projection problem in Eq. (3) and the resulted projection step in (4) is elegant and makes perfect sense to me.

**Weaknesses:**

1. More discussion on penalty and constraint-based problem formulation.

* Line 3, Page 3, I think Lisa [1] and SafeGrad [2] both explore the penalty problem formulation. I suggest the authors to add them into the citation with (Bianchi et al., 2023).

*  I can't agree the statement "Unlike penalty-based approaches, which require careful tuning of λ and offer no guarantee of feasibility, SPAG yields a closed-form projection step that enforces the constraint up to the first order."  The constraint problem and the penalty problem are sort of identical in some sense. For the constraint problem, you also need to carefully tune $\tau$, which is identical to tune $\lambda$ in the penalty problem. Please consider to remove such an unjustified statement, which is biased towards penalty methods. I tend to believe that the two methods are just two alternative way to express the same fundamental problem (trade-off between two losses), but we should not be biased towards one alternative only by simply looking at problem formulation. But I do feel that studying the constraint-based alternative is important as it gives alternative ways to design new methods and potentially these new methods can give better empirical performance.

[1] Lisa: Lazy Safety Alignment for Large Language Models against Harmful Fine-tuning Attack

2. A concurrent work should be discussed.  Stemming from the gradient conflict for the penalty problem, SafeGrad[2] derives a very similar projection method. However, because the two update rule are derived from different problems, it appears to me the fundamental projection rule seems to be not identical. I suggest the authors to:

* Discuss the similarity and difference between SafeGrad and SPAG.
* Do an experiment to compare SafeGrad and SPAG empirically. I think this is interesting because the two projection rule are derived based on different problems (penalty/constraint) and let's see which method works better empirically.

[2] SafeGrad: Gradient Surgery for Safe LLM Fine-Tuning

3. The finding of "Relevance between safety data and fine-tuning data Improves Safety" has already been covered by several existing literature but the authors did not cite and discuss them. I urge the authors to properly credit the following works, otherwise it will vitiate our research community.

* This finding is first concurrently covered by [3][4] in two ICLR2025 submissions.  Then it is also covered by [5].  Particularly, a highly relevant work [3] also discusses similarity and diversity metric. They use a similar similarity and diversity metric to measure the  a  subset of dataset.  However, this paper lacks proper credit to [3] given the relevance of these two papers. I strongly suggest the authors to properly credit [3]. Otherwise I can't recommend acceptance of this paper.   Also, could you discuss whether you have some novelty contribution over [3]? If you have, could you perform experiments to show that how your method compare against [3]?

* With that said, it seems that  the finding "Notably, the curve also shows that ASR rises again (to 16.6%) when the selected samples are too similar to the fine-tuning data " is new to me.

* A recent work [6] explores the safety sample curation problem. They explore an optimization-based solution for curate safety data. The solutions are not in the same direction with the  cosine similarity criterion explored in [3][4]. I suggest the authors compare with [6] to see which safety data curation method is better.

[3] Your Task May Vary: A Systematic Understanding of Alignment and Safety Degradation when Fine-tuning LLMs  (ICLR2025 submission)

[4] Do as I do (Safely): Mitigating Task-Specific Fine-tuning Risks in Large Language Models (ICLR2025 submission)

[5]  When Style Breaks Safety: Defending LLMs Against Superficial Style Alignment

[6] Pharmacist: Safety Alignment Data Curation for Large Language Models against Harmful Fine-tuning

4. In addition to the above highly relevant papers, there are many more papers on harmful fine-tuning that are not discussed in this paper:

Scaling Trends for Data Poisoning in LLMs

Unleashing the Unseen: Harnessing Benign Datasets for Jailbreaking Large Language Models

Virus: Harmful Fine-tuning Attack for Large Language Models Bypassing Guardrail Moderation

No, of course I can! Refusal Mechanisms Can Be Exploited Using Harmless Fine-Tuning Data

Benign Samples Matter! Fine-tuning On Outlier Benign Samples Severely Breaks Safety

Your Agent May Misevolve: Emergent Risks in Self-evolving LLM Agents

Eliciting Harmful Capabilities by Fine-Tuning on Safeguarded Outputs

Deep Ignorance: Filtering Pretraining Data Builds Tamper-Resistant Safeguards into Open-Weight LLMs

Vaccine: Perturbation-aware alignment for large language model aginst harmful fine-tuning

Tamper-Resistant Safeguards for Open-Weight LLMs

Booster: Tackling harmful fine-tuning for large language models via attenuating harmful perturbation

Targeted Vaccine: Safety Alignment for Large Language Models against Harmful Fine-Tuning via Layer-wise Perturbation

Self-Destructive Language Model

CTRAP: Embedding Collapse Trap to Safeguard Large Language Models from Harmful Fine-Tuning

Vulnerability-Aware Alignment: Mitigating Uneven Forgetting in Harmful Fine-Tuning

LoX: Low-Rank Extrapolation Robustifies LLM Safety Against Fine-tuning

Towards Resilient Safety-driven Unlearning for Diffusion Models against Downstream Fine-tuning

Antibody: Strengthening Defense Against Harmful Fine-Tuning for Large Language Models via Attenuating Harmful Gradient Influence

SEAL: Safety-enhanced Aligned LLM Fine-tuning via Bilevel Data Selection

Safety alignment should be made more than just a few tokens deep

SaLoRA: Safety-Alignment Preserved Low-Rank Adaptation

Beware of Your Po! Measuring and Mitigating AI Safety Risks in Role-Play Fine-Tuning of LLMs

Shape it Up! Restoring LLM Safety during Finetuning

Mitigating Fine-tuning Risks in LLMs via Safety-Aware Probing Optimization

Refusal-Feature-guided Teacher for Safe Finetuning via Data Filtering and Alignment Distillation

AsFT: Anchoring Safety During LLM Fine-Tuning Within Narrow Safety Basin

Defending MoE LLMs against Harmful Fine-Tuning via Safety Routing Alignment

GradShield: Alignment Preserving Finetuning

A Guardrail for Safety Preservation: When Safety-Sensitive Subspace Meets Harmful-Resistant Null-Space

Detecting Instruction Fine-tuning Attack on Language Models with Influence Function

Antidote: Post-fine-tuning safety alignment for large language models against harmful fine-tuning

Locking Down the Finetuned LLMs Safety

Panacea: Mitigating Harmful Fine-tuning for Large Language Models via Post-fine-tuning Perturbation

Safe Delta: Consistently Preserving Safety when Fine-Tuning LLMs on Diverse Datasets

Navigating the safety landscape: Measuring risks in finetuning large language models

ESTIMATING WORST-CASE FRONTIER RISKS OF OPEN-WEIGHT LLMS

Detecting Adversarial Fine-tuning with Auditing Agents

Fundamental Safety-Capability Trade-offs in Fine-tuning Large Language Models

When Style Breaks Safety: Defending Language Models Against Superficial Style Alignment

There may be more relevant works (I just list above some more recent work), and I suggest the authors to read and discuss all of the relevant works when writing the paper.

**Questions:**

1. When limiting the projection step size $\alpha$ with  (Schulman et al., 2015), will the projection still make sure that constraint in (3) strictly holds?  Did you have some results for ablation when we does not limit the step size $\alpha$ with $\eta_{safe}$?

Please address the concerns and feel free to leave me a comment in the rebuttal phase. I enjoy reading this paper overall, although I have serious concern on your Section 3.2, which do not credit properly on [3]. I will consider adjusting my score based on the rebuttal.

---

> ### Comment · Reviewer_41kS · 2025-11-19
> **Another projection method**
>
> Hi authors,
>
> I just found another ICLR2026 submission [1] talking about  a similar projection method.
>
> Your projection rule again is slightly different than them. Could you discuss  constraint-aware gradient descent [1] , SafeGrad[2] and your method to show their connection (in terms of problem formulation, transformation and the resulted projection rule)? These three methods should be highly relevant.
>
>
> [1] Security-Constrained Fine-tuning: Preventing Knowledge Restoration in Unlearned Models https://openreview.net/pdf?id=90EZvjKMqK
>
> [2]  SafeGrad: Gradient Surgery for Safe LLM Fine-Tuning

---

> ### Author Response · Authors · 2025-11-21
> **Reply to Reviewer 41kS (Part 1)**
>
> Thank you for the insightful review. We address your concerns as follows.
>
> > **W1.** Line 3, Page 3, add Lisa [1] and SafeGrad [2] into the citation with (Bianchi et al., 2023).
>
> **A1.**
> Thanks for the valuable suggestion, we have added the citation into the revision.
>
> > **W2.** I can't agree the statement "Unlike penalty-based approaches, which require careful tuning of λ and offer no guarantee of feasibility, SPAG yields a closed-form projection step that enforces the constraint up to the first order." The constraint problem and the penalty problem are sort of identical in some sense. For the constraint problem, you also need to carefully tune , which is identical to tune in the penalty problem. Please consider to remove such an unjustified statement, which is biased towards penalty methods. I tend to believe that the two methods are just two alternative way to express the same fundamental problem (trade-off between two losses), but we should not be biased towards one alternative only by simply looking at problem formulation. But I do feel that studying the constraint-based alternative is important as it gives alternative ways to design new methods and potentially these new methods can give better empirical performance.
>
> **A2.**
> We thank the reviewer for the insightful comment and agree that the previous wording overstated the distinction between constrained and penalty-based formulations. We have removed the statement accordingly. While SPARD introduces a threshold parameter $\tau$, we empirically find that SPARD is relatively insensitive to $\tau$ (Figure 4), largely because the safety loss of aligned models lies in a narrow range.
> Ultimately, we view constrained and penalty formulations as complementary perspectives, each offering distinct algorithmic opportunities for designing effective safety-preserving fine-tuning methods.

---

> ### Author Response · Authors · 2025-11-21
> **Reply to Reviewer 41kS (Part 2)**
>
> > **W3.** Discuss the similarity and difference between SafeGrad and SPAG. A concurrent work should be discussed. Stemming from the gradient conflict for the penalty problem, SafeGrad[2] derives a very similar projection method. However, because the two update rule are derived from different problems, it appears to me the fundamental projection rule seems to be not identical. I suggest the authors to: Discuss the similarity and difference between SafeGrad and SPAG. Do an experiment to compare SafeGrad and SPAG empirically. I think this is interesting because the two projection rule are derived based on different problems (penalty/constraint) and let's see which method works better empirically.
>
> **A3.**
> We thank the reviewer for raising this important point and fully agree that SafeGrad, as a concurrent work, should be discussed. We discuss SafeGard with SPARD as follows:
>
> **(i) Similarity.**
> Both SafeGrad and SPARD intervene in the gradient update to prevent harmful optimization directions. In both methods, the update rule subtracts a corrective component from the proposed step, giving them a surface-level resemblance. Conceptually, both approaches aim to preserve safety by modifying gradients rather than relying solely on loss weighting.
>
> **(ii) Difference.**
> Despite this similarity, the two methods arise from *different optimization principles*:
>
> * **SafeGrad** is derived from resolving gradient conflict in the *penalty-based* multi-objective formulation:
>   $$
>   \mathcal{L}\_{\text{ft}} + \rho \mathcal{L}\_{\text{align}}.
>   $$
> In each step, it inspects the cosine similarity between the fine-tuning and alignment gradients and modifies the fine-tuning gradient only when a gradient conflict is detected (negative cosine similarity). The projection removes the component of the fine-tuning gradient that conflicts with the alignment gradient, with the goal of avoiding destructive interference between the two losses rather than enforcing an explicit constraint.
>
> * **SPARD**, in contrast, is derived from a *safety-constrained* optimization problem:
>   $$
>   \min\_\theta \mathcal{L}\_{\text{ft}}(\theta)
>   \quad\text{s.t.}\quad
>   \mathcal{L}\_{\text{safe}}(\theta)\le\tau.
>   $$
> It first performs an unconstrained fine-tuning update and then applies a **separate safety-projection step** only when the safety loss of the updated parameters violate the threshold.
> The projection corresponds to a trust-region correction that restores feasibility to the linearized constraint boundary, rather than to the orthogonal complement of the alignment gradient.
>
> As a result, although both methods involve “subtracting a corrective component,” the **geometry, trigger conditions, and guarantees** differ significantly.
>
> **(iii) Empirical comparison.**
> Following the reviewer’s suggestion, we conducted a direct comparison under the same experimental setup. The results show that SPARD achieves substantially lower attack success rates while having a slightly better downstream performance:
>
> | Method    | BeaverTails | I-BeaverTails | LatHarmful | Q-LatHarmful | **Avg. ASR ↓** || GSM8K Acc ↑ |
> | --------- | ----------- | ------------- | ---------- | ------------ | -------------- |-|------ |
> | SafeGrad  | 32.00       | 41.72         | 20.00      | 27.00        | 30.18          || 85.71   |
> | **SPARD** | **10.60**   | **14.05**     | **6.46**   | **10.39**    | **10.38**      || **85.77**   |
>
> We appreciate the reviewer for encouraging this comparison; it has strengthened both the clarity and completeness of our submission. We have added a detailed comparison in the revised version and include the results.

---

> ### Author Response · Authors · 2025-11-21
> **Reply to Reviewer 41kS (Part 3)**
>
> > **W4.1.** Could you discuss whether you have some novelty contribution over [3]? If you have, could you perform experiments to show that how your method compare against [3]?
>
> **A4.1.**
> We thank the reviewer for the thoughtful question. Our method and [3] in fact target fundamentally different problems, and therefore their techniques and evaluation settings are not directly comparable. In addition, our selection principle and optimization method differ substantially from [3]. We provide a detailed discussion of these differences below:
>
> **(i) Different safety problem.**
> [3] investigates how to construct more durable safety guardrails **before downstream fine-tuning** by modifying the upstream safety-alignment dataset. Their goal is diagnostic: to study why guardrails collapse and how upstream diversity affects robustness.
> In contrast, SPARD focuses on a **different and complementary** problem: maintaining the safety of an already aligned model **during downstream fine-tuning itself**, independent of how the upstream dataset was created. SPARD is a *downstream defense mechanism*, whereas [3] is an *upstream alignment analysis*. Because the operational settings differ (upstream data modification vs. downstream safe optimization), a direct comparison would not be meaningful.
>
> **(ii) Different selection principles.**
> Although [3] measures similarity and diversity, their method uses only a **Top-K similarity heuristic** to select safe-alignment subsets, and **diversity is used solely as a post-hoc diagnostic metric**. The selection is not designed to support downstream safety, nor is it used as a defense.
> In contrast, SPARD introduces a **principled Relevance–Diversity DPP**, where diversity is explicitly optimized via the DPP determinant to avoid redundancy, and relevance is *modulated* to match downstream data distribution. This leads to a safe subset tailored to the downstream task—an ability that [3] does not provide.
>
> **(iii) Different mechanisms for ensuring safety.**
> Crucially, [3] performs standard fine-tuning and does not impose any optimization constraint to preserve safety. SPARD proposes a novel **Safety-Projected Alternating Gradient (SPAG)** procedure that enforces a **formal safety constraint** at every training step. This projection-based constrained optimization is unique to our method and is not present in [3].
>
> In short, [3] analyzes the *root causes* of safety degradation in upstream alignment, while SPARD provides a practical, task-aware defense for downstream fine-tuning via a new DPP selection strategy and a safety-projected optimization mechanism. Because the goals, formulations, and methodologies are fundamentally different, we believe SPARD provides substantial novelty beyond [3] and we will add this discussion to the revised paper.
>
> > **W4.2.** The finding of "Relevance between safety data and fine-tuning data Improves Safety" has already been covered by several existing literature but the authors did not cite and discuss them. This finding is first concurrently covered by [3][4] in two ICLR2025 submissions. Then it is also covered by [5]. (The finding of "Relevance between safety data and fine-tuning data Improves Safety" has already been covered by several existing literature [4][5].)
>
> **A4.2.**
> We thank the reviewer for pointing out the concurrent literature discussing the role of “relevance’’ in improving safety. We have cited and discussed these works in the revision.
>
> **(i) Relation to [4] Do As I Do (Safely).**
> This work observes that safety data that *matches the task format* of the user’s dataset improves safety during downstream fine-tuning. Their notion of relevance is therefore *template-level* or *prompt-format relevance*, and the mitigation is implemented by **rewriting safety samples** to mimic user styles before mixing them into training.
>
> **(ii) Relation to [5] When Style Breaks Safety (SafeStyle).**
> This work independently finds that safety degrades when benign fine-tuning overexposes a model to certain *style patterns*, and that injecting a small amount of safety data that matches these styles can mitigate the safety degradation. Their notion of relevance is thus *superficial style-pattern matching*.
>
> **(iii) Distinct contribution of SPARD.**
> While [4] and [5] both show that increasing *data-level or style-level similarity* between safety data and fine-tuning data can help, SPARD takes a fundamentally different and complementary perspective:
>
> 1. **Semantic relevance via embeddings, not template or style matching.** SPARD identifies safety samples whose *semantic content* aligns with harmful directions in embedding space. This differs from format-level paraphrasing [4] or style-level augmentation [5].
> 2. **Relevance–diversity principle.** SPARD shows that **relevance alone is insufficient**, and introduces a DPP-based selector that jointly maximizes semantic relevance and coverage of safety-critical behaviors, which is an aspect not addressed by [4] or [5].

---

> ### Author Response · Authors · 2025-11-21
> **Reply to Reviewer 41kS (Part 4)**
>
> > **W4.3.** A recent work [6] explores the safety sample curation problem. They explore an optimization-based solution for curate safety data. The solutions are not in the same direction with the cosine similarity criterion explored in [3][4]. I suggest the authors compare with [6] to see which safety data curation method is better.
>
> **A4.3.**
> We thank the reviewer for pointing out the recent work [6]. This work addresses **a substantially different problem setting** from ours, which makes direct comparison infeasible. Specifically, [6] focuses on *alignment-stage safety data curation* for **unaligned models**, and its optimization procedure **requires access to a harmful dataset** (used to compute harmful gradients).
> In contrast, SPARD is designed for the **downstream fine-tuning stage**, where the base model is *already safety-aligned*. SPARD does **not** rely on any harmful data, neither during selection nor during optimization. Our relevance–diversity DPP only leverages the downstream fine-tuning data together with the existing aligned safety corpus. Because the optimization signals required by [6] cannot be computed without a designated harmful dataset, their method is not applicable in SPARD’s setting.
>
> Because of this fundamental difference in assumptions, particularly the necessity of harmful data, [6] cannot be applied to SPARD’s setting, nor can SPARD be directly evaluated under their alignment-stage scenario. We have added this related work in the revised paper.
>
> > **W5.** In addition to the above highly relevant papers, there are many more papers on harmful fine-tuning that are not discussed in this paper:
>
> **A5.**
> We thank the reviewer for pointing this out. We have expanded the related work section to include additional papers on harmful fine-tuning.
>
> >**W6.** When limiting the projection step size $\alpha$ with (Schulman et al., 2015), will the projection still make sure that constraint in (3) strictly holds? Did you have some results for ablation when we does not limit the step size with $\eta\_\text{safe}$?
>
> **A6.**
> We agree with the reviewer that limiting the projection step size does **not provide a strict theoretical guarantee** that the safety constraint in Eq. (3) always holds.
> However, **empirically the constraint is consistently satisfied**.
>
> To verify this, after every projection step we recompute the safety loss under our constraint setting $(\tau=2)$. The empirical statistics across the entire training process are:
>
> | Mean | Min  | Max  |
> | ---- | ---- | ---- |
> | 0.17 | 0.02 | 0.60 |
>
> All values remain well below the threshold $(\tau=2)$, indicating that the projection keeps the model inside the safe region at every step.
>
> As suggested by the reviewer, we also conduct an ablation removing the trust-region limit $\eta\_{\text{safe}}$.
> The results below report the ASR under multiple attacks and the average downstream GSM8K accuracy.
> As can be seen, removing $\eta\_{\text{safe}}$ leads to more aggressive projection steps: safety improves on some attacks, but downstream performance degrades substantially.
>
> ||BeaverTails|I-BeaverTails|LatHarmful|Q-LatHarmful|**Avg. ASR ↓** || GSM8K Acc ↑ |
> |-|-|-|-|-|-|-|-|
> |SPARD (*w/o* $\eta\_\text{safe}$)|8.20 |10.69|0.00|1.22|5.03||81.92|
> | SPARD | 10.60   | 14.05     | 6.46   | 10.39    | 10.38      || 85.77   |
>
> These results highlight that $\eta\_{\text{safe}}$ provides a stable and balanced projection step, preserving both strong safety and good downstream performance. We have included these empirical findings in the revision.

---

> ### Author Response · Authors · 2025-11-21
> **Reply to Reviewer 41kS (Part 5)**
>
> > **W7.** Could you discuss constraint-aware gradient descent [1] , SafeGrad[2] and your method to show their connection (in terms of problem formulation, transformation and the resulted projection rule)? These three methods should be highly relevant.
>
> **A7.**
> We thank the reviewer for pointing out the connections between our method, Constraint-Aware Gradient Descent [1] (which we refer to here as CAGD), and SafeGrad [2]. We agree that these methods share a fundamental goal: reconciling utility objectives with safety requirements via gradient-based modifications.
> Below, we discuss their connections in terms of **Problem Formulation**, **Transformation**, and the **Resulted Projection Rule** as follows:
>
> **(i) Problem Formulation.** All three methods aim to optimize a utility loss $\mathcal{L}\_\text{ft}$ while managing a safety loss $\mathcal{L}\_\text{safe}$. However, they frame the optimization landscape differently:
> * **SafeGrad [2]:** Formulates the problem as **Multi-Objective Optimization** focused on **Conflict Resolution**. It does not necessarily enforce a strict loss value threshold (like $\tau$); rather, it focuses on the geometric alignment of update directions to prevent the utility gradient from increasing the safety loss.
>
> Both CAGD and SPAG formulate the task as a **Safety Constrained Optimization** problem, but they fundamentally diverge in the optimization sub-problem solved at each step:
>
> * **CAGD [1]**: Adopts a Control Barrier Function approach. It solves a **barrier constraint sub-problem** to find the minimal intervention required to ensure the trajectory remains forward-invariant within the feasible set.
> * **SPAG (Ours):** Adopts a Local Approximation approach. We solve a **feasibility restoration sub-problem** by approximating the safety constraint using a first-order Taylor expansion around the utility-updated model. This allows us to **explicitly** project parameters back into the feasible region.
>
> **(ii) Transformation and Projection Rules.**
> A fundamental distinction exists in when and how the methods apply corrections:
>
> - **SafeGrad [2] (Gradient Surgery).**
> SafeGrad checks whether the utility gradient $\textbf g\_\text{ft}$ and safety gradient $\textbf g\_\text{safe}$ conflict (i.e.,  $\textbf{g}\_\text{ft}^\top \textbf{g}\_\text{safe} < 0$).
> If they conflict, it projects $\textbf g\_\text{ft}$ onto the subspace orthogonal to $\textbf g\_\text{safe}$ and adds a scaled safety component (hyper-parameter $\rho$), i.e.,
> $$
> \begin{aligned}
> \textbf{g} &= \begin{cases}
> \textbf{g}\_\text{ft} - \frac{\textbf{g}\_\text{ft}^\top \textbf{g}\_\text{safe}}{||\textbf{g}\_\text{safe}||^2}  \textbf{g}\_\text{safe} + \rho  \textbf{g}\_\text{safe} & \text{if   } \textbf{g}\_\text{ft}^\top \textbf{g}\_\text{safe} < 0  \\\\
> \textbf{g}\_\text{ft} + \rho \textbf{g}\_\text{safe} & \text{otherwise}
> \end{cases}
> \end{aligned}
> $$
>
> - **CAGD [1] (Constraint-Aware Gradient Descent).**
> CAGD modifies the gradient flow to satisfy a barrier condition with minimal intervention.
> When the gradient alignment $(\mathbf{g}\_{\text{ft}}^\top \mathbf{g}\_{\text{safe}})$ falls below the current safety margin $\alpha(\mathcal{L}\_{\text{safe}}(\theta)-\tau)$ (a large $\alpha$), the algorithm projects $\mathbf{g}\_{\text{ft}}$ onto the subspace orthogonal to $\mathbf{g}\_{\text{safe}}$ and adds a scaled $\mathbf{g}_{\text{safe}}$ component to enforce the barrier condition:
> $$
> \begin{aligned}
> \textbf{g} &= \begin{cases}
> \textbf{g}\_\text{ft} - \frac{\textbf{g}\_\text{ft}^\top \textbf{g}\_\text{safe}}{||\textbf{g}\_\text{safe}||^2}  \textbf{g}\_\text{safe} + \frac{\alpha (\mathcal{L}\_{\text{safe}}(\theta)-\tau)}{||\mathbf{g}\_{\text{safe}}||^2}  \textbf{g}\_\text{safe} & \text{if   }   \textbf{g}\_\text{ft}^\top \textbf{g}\_\text{safe} < \alpha (\mathcal{L}\_{\text{safe}}(\theta)-\tau) \\\\
> \textbf{g}\_\text{ft}  & \text{otherwise}
> \end{cases}
> \end{aligned}
> $$
>
> - **SPAG (Safety-Projected Alternating Gradient).**
> SPAG decouples the utility update from safety enforcement through an alternating two-step procedure. We first take a utility update to obtain $\theta^+=\theta-\eta\mathbf{g}\_\text{ft}$, then evaluate the safety loss $\mathcal{L}\_\text{safe}(\theta^+)$; if it exceeds the threshold (i.e., $\mathcal{L}\_\text{safe}(\theta^+) > \tau$), we project $\theta^+$ back to the safe half-space (Line 158). The update gradient is:
> $$
> \begin{aligned}
> \textbf{g} &= \begin{cases}
> \textbf{g}\_\text{ft} + \frac{\mathcal{L}\_{\text{safe}}(\theta^{+}) - \tau}{||\mathbf{g}\_{\text{safe}}||^2}\mathbf{g}\_{\text{safe}} & \text{if   }   \mathcal{L}\_{\text{safe}}(\theta^{+}) > \tau \\\\
> \textbf{g}\_\text{ft}  & \text{otherwise}
> \end{cases}
> \end{aligned}
> $$

---

> ### Comment · Reviewer_41kS · 2025-11-21
> **Thanks for the long rebuttal**
>
> I think you still need to compare your task may vary[3] with your data selection method.
>
> You can extend [3] by first sampling safety data and then use this safety data along with the user harmful data to finetune in the user fine-tuning stage. This extension of [3] is straightforward and comparable with your data selection. Note, please disable your projection component (your first contribution) for fair comparison.
>
> Could you also compare with  CAGD, along with Safeguard, your method and SFT without defense? Also, in this experiment please disable your second component (the data selection) for fair comparison. In my understanding, the only difference of CAGD, Safeguard and your method is the constant scale factor and you should be able to do this experiment easily. To be straightforward, I want to see how the three scale factors actually affect performance.
>
> Also, for your update rule in the last comment, please modify your update rule such that the notation becomes the same with CAGD and Safeguard by clarifying the update gradient direction. It is hard to compare by looking at your formulation.

---

> > ### Author Response · Authors · 2025-11-24
> > **Reply to Reviewer 41kS (Part 7)**
> >
> > > **W10.** Also, for your update rule in the last comment, please modify your update rule such that the notation becomes the same with CAGD and Safeguard by clarifying the update gradient direction. It is hard to compare by looking at your formulation.
> >
> > **A10.**
> > We appreciate the suggestion regarding the notation. To facilitate a direct comparison with CAGD and SafeGrad, we reformulate our update rule using the unified notation and explicitly define the update gradient direction:
> >
> > - **SafeGrad [2] (Gradient Surgery).**
> > SafeGrad checks whether the utility gradient $\textbf g\_\text{ft}$ and safety gradient $\textbf g\_\text{safe}$ conflict (i.e.,  $\textbf{g}\_\text{ft}^\top \textbf{g}\_\text{safe} < 0$).
> > If they conflict, it projects $\textbf g\_\text{ft}$ onto the subspace orthogonal to $\textbf g\_\text{safe}$ and adds a scaled safety component (hyper-parameter $\rho$), i.e.,
> > $$
> > \begin{aligned}
> > \textbf{g} &= \begin{cases}
> > \textbf{g}\_\text{ft} - \frac{\textbf{g}\_\text{ft}^\top \textbf{g}\_\text{safe}}{||\textbf{g}\_\text{safe}||^2}  \textbf{g}\_\text{safe} + \rho  \textbf{g}\_\text{safe} & \text{if   } \textbf{g}\_\text{ft}^\top \textbf{g}\_\text{safe} < 0  \\\\
> > \textbf{g}\_\text{ft} + \rho \textbf{g}\_\text{safe} & \text{otherwise}
> > \end{cases}
> > \end{aligned}
> > $$
> >
> > - **CAGD [1] (Constraint-Aware Gradient Descent).**
> > CAGD modifies the gradient flow to satisfy a barrier condition with minimal intervention.
> > When the gradient alignment $(\mathbf{g}\_{\text{ft}}^\top \mathbf{g}\_{\text{safe}})$ falls below the current safety margin $\alpha(\mathcal{L}\_{\text{safe}}(\theta)-\tau)$ (a large $\alpha$), the algorithm projects $\mathbf{g}\_{\text{ft}}$ onto the subspace orthogonal to $\mathbf{g}\_{\text{safe}}$ and adds a scaled $\mathbf{g}_{\text{safe}}$ component to enforce the barrier condition:
> > $$
> > \begin{aligned}
> > \textbf{g} &= \begin{cases}
> > \textbf{g}\_\text{ft} - \frac{\textbf{g}\_\text{ft}^\top \textbf{g}\_\text{safe}}{||\textbf{g}\_\text{safe}||^2}  \textbf{g}\_\text{safe} + \frac{\alpha (\mathcal{L}\_{\text{safe}}(\theta)-\tau)}{||\mathbf{g}\_{\text{safe}}||^2}  \textbf{g}\_\text{safe} & \text{if   }   \textbf{g}\_\text{ft}^\top \textbf{g}\_\text{safe} < \alpha (\mathcal{L}\_{\text{safe}}(\theta)-\tau) \\\\
> > \textbf{g}\_\text{ft}  & \text{otherwise}
> > \end{cases}
> > \end{aligned}
> > $$
> >
> > - **SPAG (Safety-Projected Alternating Gradient).**
> > SPAG decouples the utility update from safety enforcement through an alternating two-step procedure. We first take a utility update to obtain $\theta^+=\theta-\eta\mathbf{g}\_\text{ft}$, then evaluate the safety loss $\mathcal{L}\_\text{safe}(\theta^+)$; if it exceeds the threshold (i.e., $\mathcal{L}\_\text{safe}(\theta^+) > \tau$), we project $\theta^+$ back to the safe half-space (Line 158). The update gradient is:
> > $$
> > \begin{aligned}
> > \textbf{g} &= \begin{cases}
> > \textbf{g}\_\text{ft} + \frac{\mathcal{L}\_{\text{safe}}(\theta^{+}) - \tau}{||\mathbf{g}\_{\text{safe}}||^2}\mathbf{g}\_{\text{safe}} & \text{if   }   \mathcal{L}\_{\text{safe}}(\theta^{+}) > \tau \\\\
> > \textbf{g}\_\text{ft}  & \text{otherwise}
> > \end{cases}
> > \end{aligned}
> > $$
> >
> > These formulations highlight the structural differences between the methods:
> > * **Triggering Condition (loss-based vs. gradient-based):**
> > Unlike SafeGrad and CAGD, which use **gradient alignment** ($\mathbf{g}\_{\text{ft}}^\top \mathbf{g}\_{\text{safe}}$) to activate projection,
> > our SPAG is based on **the actual safety loss** ($\mathcal{L}\_{\text{safe}}(\theta^{+}) > \tau$), providing more stable and semantically meaningful intervention.
> > Moreover, the former always requires computing $\mathbf{g}\_{\text{safe}}$ at every step, while our SPAG is **more efficient** as $\mathbf{g}\_{\text{safe}}$ is only computed when a violation occurs.
> >
> > * **Projection (gradient projection vs. parameter projection):**
> > While SafeGrad and CAGD adjust the gradient direction through projection, SPAG takes a fundamentally different approach by performing parameter-level projection (Line 158, Section 3.1), directly projecting the utility-updated model back onto the safe manifold when a violation occurs.
> >
> > Empirically, as shown in the Table in A9, SPAG delivers stronger safety than SafeGrad while preserving utility, whereas CAGD suffers a substantial drop in utility.
> > We have updated our response A7 to reflect this clarification.

---

> ### Author Response · Authors · 2025-11-24
> **Reply to Reviewer 41kS (Part 6)**
>
> > **W8.** You can extend [3] by first sampling safety data and then use this safety data along with the user harmful data to finetune in the user fine-tuning stage. This extension of [3] is straightforward and comparable with your data selection. Note, please disable your projection component (your first contribution) for fair comparison.
>
> **A8.**
> We appreciate the suggestion to isolate the contribution of our data selection strategy. Following your recommendation, we conducted an additional experiment extending [3] by sampling safety data and using it for fine-tuning without our projection component (SPAG).
>
> **1. Performance without SPAG (Data Selection Only).**
> We compared our **Relevance-Diversity DPP** selection against the strategies from [3] ($\mathcal D_\text{Low-Sim}$ and $\mathcal D_\text{High-Sim}$) using standard fine-tuning. As shown in the table belows, the average ASR remains high across the board (mostly $>73\%$) when SPAG is disabled, and the performance gap between different data selection methods is marginal.
>
> This highlights the importance of an explicit safety constraint: without it, even well-selected safety data (e.g., $\mathcal D_\text{High-Sim}$, or our DPP selection) cannot fully realize its potential to counteract the harmful fine-tuning data.
>
> | Method    | BeaverTails | I-BeaverTails | LatHarmful | Q-LatHarmful | **Avg. ASR ↓** || GSM8K Acc ↑ |
> | --------- | ----------- | ------------- | ---------- | ------------ | -------------- |-|------ |
> | $\mathcal D_\text{Low-Sim}$ [3]  | 74.60   | 70.65   | 88.48 | 94.70 | 82.11         ||  82.11  |
> | $\mathcal D_\text{High-Sim}$ [3]  | 67.80  | 69.39   | 74.95      | 86.35        | 74.62          ||  86.62  |
> | Relevance-Diversity DPP | 70.40   | 66.46    | 73.33   | 89.41    | 74.9     || 86.01   |
>
> **2. Performance with SPAG.**
> To meaningfully distinguish the effectiveness of different data selection strategies,
> we further examine all selection strategies when combined with **SPAG**, i.e., with the safety constraint enforced during fine-tuning.
>
> As shown in the Table below, our method achieves the lowest average ASR performance. Compared with the high-similarity set $\mathcal D_\text{High-Sim}$ [3], our approach reduces the average ASR from 16.51% to **10.38%** while maintaining comparable utility on GSM8K.
> Meanwhile, the low-similarity set $\mathcal D_\text{Low-Sim}$ [3] performs substantially worse, confirming that such low-similarity safety samples are much less useful for enforcing the safety constraint.
> These results demonstrate that our Relevance-Diversity DPP selection is more effective at selecting safety data than the relevance-only metrics in [3].
>
> | Method    | BeaverTails | I-BeaverTails | LatHarmful | Q-LatHarmful | **Avg. ASR ↓** || GSM8K Acc ↑ |
> | --------- | ----------- | ------------- | ---------- | ------------ | -------------- |-|------ |
> | SPAG (w/ $\mathcal D_\text{Low-Sim}$[3])  | 34.80   | 56.39   | 64.24      | 79.84        | 58.82          ||  84.95  |
> | SPAG (w/ $\mathcal D_\text{High-Sim}$[3])  | 16.60   | 24.32   | 16.16      | 8.96        | 16.51          ||  85.69  |
> | SPARD | 10.60   | 14.05     | 6.46   | 10.39    | 10.38      || 85.77   |
>
> > **W9.** Could you also compare with CAGD, along with Safeguard, your method and SFT without defense? Also, in this experiment please disable your second component (the data selection) for fair comparison. In my understanding, the only difference of CAGD, Safeguard and your method is the constant scale factor and you should be able to do this experiment easily. To be straightforward, I want to see how the three scale factors actually affect performance.
>
> **A9.**
> We appreciate the reviewer’s suggestion. We conducted additional experiments to directly compare SPAG against CAGD and SafeGrad using the entire safety dataset (without data selection) following the setting in SafeGrad.
>
> As shown in the table below, SPAG achieves an effective balance between safety and utility.
> Specifically, SPAG achieves a substantially lower ASR (-13.34\%) than SafeGrad while maintaining comparable accuracy on GSM8K. While CAGD also attains high safety, it suffers a severe decline in utility, dropping even below the pre-trained model’s baseline accuracy (77.71%).
>
> | Method    | BeaverTails | I-BeaverTails | LatHarmful | Q-LatHarmful | **Avg. ASR ↓** || GSM8K Acc ↑ |
> | --------- | ----------- | ------------- | ---------- | ------------ | -------------- |-|------ |
> | CADG  | 8.80       | 9.01   | 0      | 0.61        | 4.61          ||  65.13  |
> | SafeGrad  | 32.00       | 41.72         | 20.00      | 27.00        | 30.18          || 85.71   |
> | **SPAG** | 19.00   | 28.30     | 12.93   | 7.13    | 16.84      || 85.03   |

---

> ### Comment · Reviewer_41kS · 2025-11-24
> **The two results are nice and clear**
>
> I think the two new results A9 and A8 clear up my concern, which clearly show the advantage of the two components (projection and safety data selection) of SPAG.
>
> One more revision suggestion:  please modify the manuscript by adding discussion with You task may vary[3] and Pharmacist[6]  in the beginning of yoru Section 3.2. [3] and your method both consider data similarity between safety data and fine-tuning data. You should definietely give this credit to [3]. On the other hand, although I am aware that [3] do not add the diveristy into the selection objective, they do have some analysis with this diversity metric.  Please avoid to say that because you are considering safety data curation in the fine-tuning stage, but they are considering safety data curation in the alignment stage, the two methods are different and therefore you don't need to credit them. This statement is aparrantly problematic.  Please also update your comparison result with [3] into the manuscript, consider the simialrity of the two methods.
>
> Also, please update your comparison with safegrad and CAGD into appendix. The results are very interesting, at least to me.

---

> ### Author Response · Authors · 2025-11-25
> **Reply to Reviewer 41kS (Part 8)**
>
> Thank you for the further comments.
> We are glad that our reply and additional experiments have resolved your previous concerns, and sincerely thank you for the constructive suggestions.
>
> > **W10.** One more revision suggestion: please modify the manuscript by adding discussion with You task may vary[3] and Pharmacist[6] in the beginning of yoru Section 3.2. [3] and your method both consider data similarity between safety data and fine-tuning data. You should definietely give this credit to [3]. On the other hand, although I am aware that [3] do not add the diveristy into the selection objective, they do have some analysis with this diversity metric. Please avoid to say that because you are considering safety data curation in the fine-tuning stage, but they are considering safety data curation in the alignment stage, the two methods are different and therefore you don't need to credit them. This statement is aparrantly problematic. Please also update your comparison result with [3] into the manuscript, consider the simialrity of the two methods.
>
> **A10.**
> Following your suggestions, we have revised Section 3.2 to explicitly discuss [3] and [6], and credited them.
>
> **Revision in Section 3.2:**
> - **Lines 200-204**: "Recent studies have also observed that relevant safety data can improve safety during training by leveraging embedding similarity (Hsiung et al., 2025), employing trained selectors (Liu et al., 2025b), or matching task styles and formats (Eiras et al., 2024; Xiao et al.,
> 2025)."
> - **Line 232**: "While Hsiung et al. (2025) introduces a metric for assessing subset diversity, they do not integrate this metric into the selection process itself, so their method cannot promote diversity during data selection."
>
> **Additional Results:** We have added the performance comparison with the selection strategies from [3] (specifically their $\mathcal{D}\_{\text{Low-Sim}}$ and $\mathcal{D}\_{\text{High-Sim}}$) to Appendix E.2 (Tables 6-7).
>
> > **W11.** Also, please update your comparison with safegrad and CAGD into appendix. The results are very interesting, at least to me.
>
> **A11.**
> We thank the reviewer for the positive feedback on these additional experiments. Following your suggestion, we have included the empirical comparison with SafeGrad and CAGD in Appendix E.3 of the revised manuscript.

---

> ### Comment · Reviewer_41kS · 2025-11-25
> **The reference in appenix e.2 is wrong**
>
> The table 6 and 7 in Appendix E.2 refer to the wrong reference. It shoud be refer to [3]. Please check. Also please add a sentence to refer to this comparison reuslts in the main paper, e.g., [3] also explore similarity and diversity metric in safety data curation. we add more discussion on the difference and similarity in appendix e.2 and present comparison results of the two methods.

---

> > ### Author Response · Authors · 2025-11-25
> > **Reply to Reviewer 41kS (Part 9)**
> >
> > We thank you for pointing out the typos; we have corrected them accordingly.
> >
> > Following your constructive suggestion, we have revised Section 4.4 (Lines 462-465) to refer to the comparison with [3].
> >
> > **Revision in Section 4.4:**
> > "Moreover, as Hsiung et al. (2025) also explores similarity and diversity metrics in safety data curation, we provide further discussion on the similarities and differences, along with an empirical comparison of the two methods, in Appendix E.2."

---

> > > ### Comment · Reviewer_41kS · 2025-11-26
> > > **Score increase**
> > >
> > > Thanks for the revision. I have updated my score to 8. I will also be happy to champion this paper in review process. Feel free to ping me if you want me to participate in the other reviewer's rebuttal.

---

> ### Author Response · Authors · 2025-11-26
> **Thank You for Raising the Score to 8!**
>
> Dear Reviewer 41kS,
>
> We sincerely appreciate your constructive feedback, which has significantly improved our work.
>
> Thank you for **raising the score to 8** and for your strong support of our paper. We are grateful for your engagement during the discussion phase and for **championing our work**.
>
> Best Regards,
>
> The Authors

---

### Official Review · Reviewer_pm8L · 2025-10-29

**Soundness:** 3
**Presentation:** 3
**Contribution:** 3
**Rating:** 6
**Confidence:** 3

**Summary:**

This paper presents a framework named SPARD to defend LLMs against harmful fine-tuning attacks. The proposed solution consists of two key components: (1) Safety-Projected Alternating Gradient (SPAG). This is a principled optimization strategy. Instead of using safety data as a soft penalty (which is common in other methods), SPAG formulates the problem as a safety-constrained optimization. It alternates between a standard utility update on the fine-tuning data and an explicit, closed-form projection step that pushes the model parameters back into a safe region defined by a curated set of safe data. (2) Relevance-Diversity Data Selection: the authors show that safety data must be both relevant to the downstream task and diverse. To achieve this balance, they propose a Relevance-Diversity Determinantal Point Process (DPP) to incorporate both a task-relevance quality score and an intrinsic diversity measure. Experiments conducted on two LLMs (Qwen-2.5-7B and LLaMA-3.2-3B) , two downstream tasks (GSM8K and OpenBookQA) , and four different harmful attack datasets demonstrate that SPARD consistently achieves the lowest ASR and HS. Notably, it does this while maintaining high task accuracy.

**Strengths:**

- The combination of SPAG and Relevance–Diversity DPP is a novel and principled solution to address harmful fine-tuning attacks.
- SPARD consistently outperforms baseline methods across multiple datasets and model architectures.
- The paper provides extensive experiments, sensitivity analyses, and comparisons with baselines, demonstrating the effectiveness and generalizability of SPARD.

**Weaknesses:**

- [Minor] Details on generating the embeddings are not directly available until Section 4.1, making people slightly confused at the beginning.
- Would the cost of applying DPP increase with the dataset size as it seems to compute pairwise similarity? how would it scale to larger datasets if that's the case?
- Computational cost: if the threshold in Algorithm 1 is passed, then a second backward is required. This will impose greater computational cost and longer running time. It would be great if the authors can provide statistics on how frequent this will trigger.
- The method's success appears to be highly dependent on several new hyperparameters that must be carefully tuned, but the paper provides limited guidance on how to set them universally.

**Questions:**

See above.

---

> ### Author Response · Authors · 2025-11-21
> **Reply to Reviewer pm8L (Part 1)**
>
> Thank you for the insightful review. We address your concerns as follows.
>
> > **W1.** Details on generating the embeddings are not directly available until Section 4.1, making people slightly confused at the beginning.
>
> **A1.**
> We thank the reviewer for pointing out this clarity issue. Following the common mean-pooling strategy [1], we generate embeddings by averaging the final-layer hidden states across all tokens in the input sequence.
> Specifically, let $\phi\_t(\mathbf{x})$ denote the hidden state at position $t$ for the input sequence $\mathbf{x} = (x\_1, \ldots, x\_T)$.
> The embedding is computed as:
> $$
> \phi(\mathbf{x}) = \frac{1}{T} \sum\_{t=1}^{T} \phi\_t(\mathbf{x}).
> $$
> This simple pooling strategy is widely used and shown to be effective for LLM-based embedding extraction.
> We have added this clarification to the paper for improved readability.
>
> [1] *Repetition Improves Language Model Embeddings*, ICLR 2025.

---

> ### Author Response · Authors · 2025-11-21
> **Reply to Reviewer pm8L (Part 2)**
>
> > **W2.** Would the cost of applying DPP increase with the dataset size as it seems to compute pairwise similarity? how would it scale to larger datasets if that's the case?
>
>
> **A2.**
> **Yes**, the cost of applying DPP increases with the size of the safety pool, but in practice it is negligible (≈0.1 seconds) compared to the overall training time (≈4 hours).
> The main computational overhead arises from the **greedy DPP selection**, *not* from computing pairwise similarities. The similarity matrix is computed **once**, entirely on GPU, and does *not* dominate the runtime.
> Below we provide both a theoretical analysis and empirical results of how this selection procedure scales with dataset size.
>
> **(i) Theoretical Analysis.**
> Let $N = \lvert \mathcal D\_{\text{safe}} \rvert$ be the size of the safe pool and let a small $k = \lvert \mathcal C \rvert$ be the target number of selected samples.
> In our experiments, $N \approx 17{,}000$ and $k = 236$, so $k \ll N$, i.e., only a small fraction of the safe pool is selected.
>
> At greedy step $m$ (with $m-1$ items already selected), we maintain the Cholesky factor $\widehat{\mathbf L}\_{\mathcal C\_{m-1}} = \mathbf C \mathbf C^\top$, where $\mathbf C \in \mathbb{R}^{(m-1)\times (m-1)}$.
> To evaluate the gain factor for all remaining candidates $i \notin \mathcal C\_{m-1}$, we first extract the cross-kernel block
> $$
>     \mathbf V\_{m-1}
>     = \widehat{\mathbf L}\_{\mathcal D\_{\text{safe}}\setminus \mathcal C\_{m-1}, \mathcal C\_{m-1}}
>     \in \mathbb{R}^{(N-m+1)\times (m-1)},
> $$
> and then solve the triangular system
> $$
>     \mathbf C \mathbf W\_{m-1}^\top = \mathbf V\_{m-1}^\top,
> $$
> where each column of $\mathbf W\_{m-1}$ corresponds to the vector $\mathbf w\_i$ used in the gain $\widehat{\mathbf L}\_{ii} - \\|\mathbf w\_i\\|\_2^2$.
>
> Following [1], we avoid repeatedly solving triangular systems from scratch.
> Instead, for each candidate item $i$, we maintain its Cholesky coordinates and residual
> $$
>     \mathbf{w}\_i \in \mathbb{R}^{m-1},
>     \qquad
>     d\_i^2 = \widehat{\mathbf{L}}\_{ii} - \lVert \mathbf{w}\_i \rVert\_2^2,
> $$
> and update them *incrementally* when a new element $j$ is added to $\mathcal{C}\_{m-1}$.
> The update for each remaining candidate item $i$ is
> $$
>     e\_i
>     = \frac{
>         \widehat{\mathbf{L}}\_{ij} -
>         \langle \mathbf{w}\_i,\, \mathbf{w}\_j \rangle
>       }{
>         d\_j
>       },
>     \qquad
>     \mathbf{w}\_i \leftarrow [\mathbf{w}\_i,\, e\_i],
>     \qquad
>     d\_i^2 \leftarrow d\_i^2 - e\_i^2,
> $$
> which requires only an inner product of length $m-1$.
> Thus, each candidate update costs $\mathcal{O}(m)$, and the entire gain update at step $m$ costs
> $$
>     \mathcal{O}\left((N - m + 1)\, m\right).
> $$
>
> Summing over all greedy steps $m = 1,\dots,k$ yields
> $$
>     \sum\_{m=1}^{k} \mathcal{O}\left((N-m+1)\, m\right)
>     = \mathcal{O}(N k^2)
>     \qquad (N \gg k).
> $$
> Thus, the final time complexity is $\mathcal{O}(N k^2)$, i.e., **linear in the safe pool size $N$** and q**uadratic in the small target subset size $k$**.
>
> **(ii) Emipirical Results.**
>
> We benchmark the Relevance–Diversity DPP selection on a single NVIDIA A800 GPU using safe datasets of varying sizes.
> As shown in the table below, the runtime is **consistently negligible**. It increases **almost linearly** with the number of safe candidates (as the $\mathcal{O}(N)$ analysis), but remains under 0.5 seconds even when selecting from 70K candidate samples.
>
> | # Safe Pool Samples   | 10K  | 30K  | 50K  | 70K  |
> |------------------|------|------|------|------|
> | Times (second)   | 0.08  | 0.14 | 0.25 | 0.46|
>
> We also evaluate the effect of varying the number of selected samples $k$.
> As shown in the table below, the runtime remains **negligible**, staying under 3 seconds even when selecting several thousand samples.
> Crucially, in our experiments we select a subset of safe samples whose size is only 3% of the fine-tuning dataset, so the total overhead remains **very small (≈0.1 s) compared to the full training time (≈4 hours)**.
>
> | # Selected samples  |0.25K|0.5K|1K|2K|3K|4K|
> |------------------|------|------|------|------|---|-|
> | Times (second)   | 0.12|0.20|0.41|0.99|1.79|2.74|
>
> [1] Fast greedy map inference for determinantal point process to improve recommendation diversity, NeurIPS 2018.

---

> ### Author Response · Authors · 2025-11-21
> **Reply to Reviewer pm8L (Part 3)**
>
> > **W3.** Computational cost: if the threshold in Algorithm 1 is passed, then a second backward is required. This will impose greater computational cost and longer running time. It would be great if the authors can provide statistics on how frequent this will trigger.
>
> **A3.**
> Thanks for your suggestion.
> We provide statistics on how often the projection threshold in Algorithm 1 is activated, which determines when an additional backward pass is required.
> We divide the entire training process into 10 equal step ranges and report the percentage of steps in each range where the projection is triggered:
>
> | **Step Scope** | 0–61  | 62–123 | 124–185 | 186–247 | 248–309 | 310–371 | 372–433 | 434–495 | 496–557 | 558–620 |
> | -------------- | ----- | ------ | ------- | ------- | ------- | ------- | ------- | ------- | ------- | ------- |
> | **Trigger %**  | 98.38 | 96.77  | 96.77   | 93.54   | 91.93   | 82.25   | 70.96   | 67.74   | 58.06   | 62.90   |
>
> As can be seen, the projection is triggered very frequently at early training stages (≈98%), indicating that the model initially violates safety constraints more often. As training progresses, the trigger rate steadily decreases to ≈58–63%, showing that safety alignment becomes easier and the additional backward pass becomes less frequent.
> This pattern demonstrates that SPAG triggers the safety-related backward pass only when necessary, which helps maintain overall training efficiency.
>
> This provides a concrete estimate of the computational overhead: although the projection is frequently triggered early on, its cost gradually decreases as training stabilizes.
>
>
> > **W4.** The method's success appears to be highly dependent on several new hyperparameters that must be carefully tuned, but the paper provides limited guidance on how to set them universally.
>
> We appreciate the reviewer’s concern regarding the sensitivity of the newly introduced hyperparameters. In practice, SPARD is **not sensitive** to these hyperparameters, and we found that simple default choices work reliably across all datasets.
>
> * **Safe sample ratio $p$.**
> As shown in Figure 3, a very small $p$ provide insufficient coverage, whereas a very large $p$ introduces redundancy. We recommend setting $p \in [0.03, 0.1]$, which consistently yields strong performance.
>
> * **Safety threshold $\tau$.**
>   Figure 4 shows that SPARD is *not sensitive* to $\tau$ when it is set to a relatively low value. We therefore recommend choosing $\tau < 3$, or initializing $\tau$ by referencing the safety loss of the pretrained aligned model on the safe dataset.
>
> * **Relevance exponent $\beta$.**
>   $\beta$ controls how strongly relevance affects DPP selection. As illustrated in Figure 5, performance remains **stable** for $\beta \in [4, 10]$. We adopt values within this range across all experiments without dataset-specific tuning.
>
> * **Trust-region radius $\eta\_{\text{safe}}$.**
>   $\eta\_{\text{safe}}$ does not require special tuning; it is sufficient to set it equal to the standard learning rate $\eta$ used for fine-tuning. This ensures the safety projection step moves at a comparable scale to the main update.
>
> Overall, SPARD achieves strong and consistent performance with these straightforward defaults, and does not require extensive hyperparameter search. We have clarified these recommendations in the revised manuscript.

---

> > ### Comment · Reviewer_pm8L · 2025-11-25
> >
> > Thank you for your clarification. I don't have any further question.

---

> > > ### Author Response · Authors · 2025-11-25
> > > **Thank you for the positive score!**
> > >
> > > Dear Reviewer pm8L,
> > >
> > > Thank you for your follow-up comments.
> > >
> > > We are glad that your concerns have been resolved and that you maintained a positive score.
> > >
> > > Best regards,
> > >
> > > The Authors

---

### Official Review · Reviewer_BprZ · 2025-10-31

**Soundness:** 2
**Presentation:** 2
**Contribution:** 2
**Rating:** 2
**Confidence:** 4

**Summary:**

This paper presents SPARD, a safety projection approach based on relevance-based diversity-aware data selection. The  relevance-based diversity-aware data selection is directly achieved by using the existing DDPs approach (Determinantal Point Processes) which ranks the top subset that contains both individually informative and mutually diverse data items/elements.

**Strengths:**

The idea of utilizing the existing DDPs approach to provide the relevance-based diversity-aware data selection is simple and interesting.

The safety projected alternating gradient (SPAG) is an extension of Bianchi et al 2023's approach, aiming to improve the tuning efforts of setting the average safety loss parameter and the penalty parameter.

The idea of extension is to perform fine-tuning on the fine-tuning dataset first (i.e., utility driven update) and then perform projection to encode the updated parameters into the half-space defined by C^+, which satisfies safety constraints.

The paper provides experimental comparison using two LLMs: Qwen-2.5-7B-instruct and LLaMA-3.2-3B-instruct and compared to three existing safety guardrail methods in addition to the SFT baseline.

**Weaknesses:**

The paper can benefit from providing clear elaboration on the following aspects.

(1) Although the idea of utilizing the existing DDPs approach to provide the relevance-based diversity-aware data selection is simple and interesting, given that the quality of selected subsets by DPPs is based on the balancing of the two criteria: individually informative and mutually diverse, the paper should provide discussion to elaborate on how these two criteria are semantically measured since informative is context relevant and mutually diverse is also context driven (e.g., agreement diversity or disagreement diversity ... ) and why DDPs based  data selection will help mitigating harmful fine-tuning.

(2) The proposed approach relies on task-relevant safe samples to provide safety constraints. It is a very strong assumption. A discussion on how the proposed approach responds when the task relevant safe samples are of varying quality and volume.

(3) The experimental results could benefit from more detailed discussion to elaborate on the boundary cases. For example, Table 1 has shown that there are two out of four datasets the proposed approach did not outperform existing safety guardrail methods, like LISA. The intuitions behind the proposed approach vs LISA (Huang et. al. 2024b) and vs. SafeInstr (Bianchi et.al 2023) should be provided.

(4) Similar questions also apply to Table 2, Table 3 and Table 4.

(5) Can you use Figure 6 style of comparison on GSM8K to show those cases where SPARD performance is weaker compared to existing methods and analyze why.

**Questions:**

See the weakness section

---

> ### Author Response · Authors · 2025-11-21
> **Reply to Reviewer BprZ (Part 1)**
>
> Thank you for the insightful review. We address your concerns as follows.
>
> > **W1.** Although the idea of utilizing the existing DDPs approach to provide the relevance-based diversity-aware data selection is simple and interesting, given that the quality of selected subsets by DPPs is based on the balancing of the two criteria: individually informative and mutually diverse, the paper should provide discussion to elaborate on how these two criteria are semantically measured since informative is context relevant and mutually diverse is also context driven (e.g., agreement diversity or disagreement diversity ... ) and why DDPs based data selection will help mitigating harmful fine-tuning.
>
> **A1.**
> We thank the reviewer for raising this important point regarding the semantic meaning of *relevance* and *diversity* in our DPP-based selection.
> Below, we clarify how each component is defined, why these definitions are meaningful in the context of safety alignment, and why Relevance–Diversity DPP are an appropriate mechanism for balancing them.
>
> **(1) How we measure *relevance.***
>
> Relevance aims to ensure that selected safe samples provide strong corrective signals for the harmful behaviors present in the fine-tuning data.
> As described in Lines 211-222 of Section 3.2, we embed every safe prompt using the pretrained encoder and compute its cosine similarity against all fine-tuning data's embeddings.
> For each safe sample, **the maximum cosine similarity** serves as its quality score, capturing how closely it aligns with the most similar fine-tuning instance.
> As shown in Figure 2 of the paper, higher relevance (i.e., greater similarity between safe and fine-tuning embeddings) significantly reduces attack success rate. This aligns with the intuition that safety constraints are most effective when they directly counteract the directions along which harmful fine-tuning drifts the model.
>
>
> **(2) How we measure *diversity.***
>
> While high relevance is beneficial, selecting overly similar safe samples leads to redundancy (Figure 2) and weak coverage of diverse harmful modes (Figure 6). Therefore, we incorporate **distributional diversity** among safe samples.
>
> As detailed in Lines 237–254 of Section 3.2, to operationalize this, we compute pairwise cosine similarities between safe-sample embeddings and use them to form a kernel matrix. Diversity is then quantified through the **determinant** of the DPP kernel [1], which is well-established [1-2] as a principled measure of geometric volume. Intuitively, a larger determinant encourages selecting samples that span a larger subspace, i.e., samples that are *mutually dissimilar*.
>
> [1] Determinantal Point Processes for Machine Learning, Foundations and Trends in Machine Learning.
>
> [2] Determinantal point process models and statistical inference: Extended version, Journal of the Royal Statistical Society Series B.
>
> **(3) Why DPP-based relevance–diversity selection mitigates harmful fine-tuning**
>
> Accordingly, an effective safety set for counteracting harmful fine-tuning should:
>
> 1. **Match** the harmful directions (high relevance), and
> 2. **Cover** the full spectrum of potential safety violations (high diversity).
>
> Our DPP formulation is particularly suitable because it *naturally encodes both terms*: the determinant structure encourages diversity and a relevance weight that biases selection toward fine-tuning-aligned safe samples.
>
> Empirically, we observe precisely this behavior:
>
> * Relevance alone reduces ASR but but over-emphasizing relevance leads to degraded performance (Figure. 2).
> * Diversity alone avoids redundancy but lacks direct corrective impact. (Table 4).
> * Combining both via a relevance-modulated DPP yields the strongest mitigation (Figure. 6, t-SNE visualizations).

---

> ### Author Response · Authors · 2025-11-21
> **Reply to Reviewer BprZ (Part 2)**
>
> > **W2.** The proposed approach relies on task-relevant safe samples to provide safety constraints. It is a very strong assumption. A discussion on how the proposed approach responds when the task relevant safe samples are of varying quality and volume.
>
> **A2.**
> We respectfully clarify that our method **does not rely on perfectly task-aligned or high-quality safe samples**, nor does it require a large number of them.
> In practice, SPARD only assumes access to some safe data, which is a standard requirement also shared by existing safety fine-tuning and preference-based alignment methods.
> Importantly, all methods in our comparison (PTST, SafeInstr, Lisa) are trained with the same safe dataset, yet SPARD consistently achieves superior safety.
>
> To directly evaluate robustness under varying task relevance, we further test SPARD on six additional attack tasks (AdvBench, I-CoNa, I-Controversial, I-MaliciousInstructions, I-PhysicalSafetyUnsafe, and Q-Harm) [1-2].
> These attacks **differ substantially** from the corpora used to build our candidate safety pool **GeneralSafe**, spanning different domains, harmful behavior types, and linguistic structures.
> This creates challenging settings where the quality and relevance of available safe samples vary widely across attacks.
>
> As shown in the table below, SPARD achieves the lowest average ASR across all attacks, substantially outperforming all baselines while maintaining competitive utility on GSM8K. The consistent gains across diverse attack distributions demonstrate that SPARD is robust to various tasks and does not rely on unrealistically strong assumptions about safe-sample relevance.
>
> | Model       | AdvBench | I-CoNa  | I-Controversial | I-MaliciousInstructions | I-PhysicalSafetyUnsafe | Q-Harm | **Avg. ASR ↓** || GSM8K Acc ↑ |
> |-------------|----------|---------|------------------|--------------------------|--------------------------|--------|--------|-|----------|
> | Qwen2.5-7B  | **2.00** | 17.98   | 20.00            | 9.00                     | 33.00                    | 27.00  | 18.16   || 77.71     |
> ||
> | SFT         | 83.50    | 69.10   | 57.50            | 73.25                    | 70.50                    | 74.50  | 71.39   || **86.77** |
> | PTST        | 61.80    | 45.09   | 39.38            | 59.00                    | 56.75                    | 53.00  | 52.50   || 85.06     |
> | SafeInstr   | 43.15    | 50.00   | 44.38            | 45.50                    | 58.50                    | 57.25  | 49.80   || 86.28     |
> | Lisa        | 4.10     | 19.38   | 16.25            | 10.25                    | 36.00                    | 26.50  | 18.75   || 78.45     |
> | SPARD   | 2.05     | **11.80** | **7.50**        | **3.25**                 | **17.25**                | **17.00** | **9.81** || 85.77     |
>
> We have clarified this point in the revised version and explicitly emphasize that SPARD’s effectiveness is validated under multiple distinct attack datasets, showing that it can defend against different harmful behaviors without requiring unrealistically strong assumptions on the safe sample pool.
>
> [1] Universal and Transferable Adversarial Attacks on Aligned Language Models, arXiv.
>
> [2] Safety-Tuned LLaMAs: Lessons From Improving the Safety of Large Language Models that Follow Instructions, ICLR 2024.

---

> ### Author Response · Authors · 2025-11-21
> **Reply to Reviewer BprZ (Part 3)**
>
> > **W3** The experimental results could benefit from more detailed discussion to elaborate on the boundary cases. For example, Table 1 has shown that there are two out of four datasets the proposed approach did not outperform existing safety guardrail methods, like Lisa. The intuitions behind the proposed approach vs Lisa (Huang et. al. 2024b) and vs. SafeInstr (Bianchi et.al 2023) should be provided. Similar questions also apply to Table 2, Table 3 and Table 4.
>
> **A3.**
> We thank the reviewer for pointing out the importance of analyzing boundary cases and providing more intuition on how SPARD compares to **Lisa** and **SafeInstr**. Our main goal is to optimize the **safety–utility trade-off**, rather than safety in isolation, and the per-dataset results in Tables 1–4 reflect different points on this trade-off **pareto frontier**.
>
> 1. **Comparison to Lisa (safety vs. utility).**
> Lisa achieves good safety performance on some benchmarks, but this comes at a substantial cost in task utility. In particular, Lisa often leads to markedly lower accuracy on downstream tasks (e.g., GSM8K and OpenBookQA in Tables 1–4), whereas SPARD preserves performance much closer to the SFT fine-tuning baselines.
>    Intuitively, Lisa applies a strong global safety regularization that can over-constrain the model, leading to conservative behavior and loss of general capabilities. In contrast, SPARD enforces **safety constraints** only through the selected safe samples and a safety threshold ($\tau$). This keeps the safety loss under control while still allowing the model to adapt to the fine-tuning data, which explains why SPARD typically achieves **comparable or better safety than Lisa, but with much better utility**.
>
> 2. **Comparison to SafeInstr (utility vs. safety).**
>    SafeInstr, conversely, tends to preserve or even slightly improve task utility compared to SFT, but its **safety performance is consistently weaker** than SPARD across the attack benchmarks. In Tables 1-3, SafeInstr’s ASR is substantially higher than SPARD’s on almost all attack datasets.
>    This is aligned with the design of SafeInstr: it only adds randomly selected safe samples into standard fine-tuning, without explicitly optimizing against the specific harmful behaviors induced by the fine-tuning data. SPARD, on the other hand, uses **relevance–diversity DPP selection** to pick safe samples that closely match the harmful directions while covering diverse violation modes, and then directly constrains the safety loss during fine-tuning. As a result, SPARD achieves **much lower ASR** than SafeInstr while retaining comparable utility.
>
> 3. **Boundary cases in Tables 1–4.**
> Across boundary cases in these tables, we observe SPARD typically lies the **Pareto frontier** of safety and utility, whereas other methods tend to optimize only one side of the trade-off:
>
>    * When a baseline (Lisa) achieves slightly lower ASR than SPARD on a particular dataset, it does so at the expense of a **large drop in utility**, whereas SPARD remains close to the base model’s performance.
>    * When a baseline (SafeInstr) has utility on par with or slightly above SPARD, its **ASR is markedly higher**, indicating insufficient protection against certain attack types.
>
> In the revised version, we explicitly highlight these boundary cases and emphasize that SPARD is achieving a **balanced and robust safety–utility trade-off** across diverse attack datasets.

---

> ### Author Response · Authors · 2025-11-21
> **Reply to Reviewer BprZ (Part 4)**
>
> > W4. Can you use Figure 6 style of comparison on GSM8K to show those cases where SPARD performance is weaker compared to existing methods and analyze why.
>
> **A4.**
> We thank the reviewer for the helpful suggestion. Following the visualization style of Figure 6, we provide a t-SNE comparison for the **Q-LatHarmful** setting in **Figure 8** of the Appendix. This is the scenario where Lisa slightly outperforms SPARD on safety. As shown in the visualization, Lisa uses the entire safety dataset, whereas SPARD operates with only a small relevance–diversity-selected subset. Importantly, we do **not** attribute the gap on this particular case to data selection. If the missing samples were the primary cause, we would expect a clear “coverage deficit” pattern in the embedding space, but the t-SNE plots show that SPARD’s selected points still cover the harmful region well.
>
> Instead, the difference is better explained by the **optimization principle**.
> Lisa employs a global safety regularization term, which aggressively pulls the model toward the safe region. This can yield lower ASR on some adversarial distributions, but it also **over-constrains the model**, which we observe as **significantly weaker utility performance** on GSM8K(-8.3%) and OpenBookQA(-7.2%).
>
> In contrast, **SPARD enforces safety *only when needed***: the projection step is triggered adaptively when the safety constraint is at risk of being violated. This targeted adjustment avoids unnecessary distortion of the task-learning gradient, enabling SPARD to maintain a **more favorable safety–utility balance** across datasets.
> We have added this discussion and the visualization to the Appendix.

---

> > ### Comment · Reviewer_BprZ · 2025-11-25
> >
> > I appreciate the clarification from the authors and will keep my current scores.

---

> ### Author Response · Authors · 2025-11-25
> **Request for Specific Feedback**
>
> Dear Reviewer BprZ,
>
> Thank you for your follow-up comment.
>
> We are glad our clarifications have addressed your earlier concerns.
>
> Given your current score of 2 (“reject”), **we kindly ask if you could reconsider and raise your score, or let us know any remaining issues that prevent doing so**.
>
> Best regards,
>
> The Authors

---

> > ### Comment · Reviewer_BprZ · 2025-11-26
> >
> > Unfortunately, you have not addressed all concerns raised.
> >
> > I only confirmed that I read your clarification, especially on promoting the strength of your approach. There are also recent work on harmful fine-tuning, including ICML 2025, ICLR 2025. Your comparison to only a very small number of defense methods. LISA appeared in 2024 and most methods in 2025 have shown better defense than LISA.

---

> > > ### Comment · Reviewer_BprZ · 2025-11-26
> > >
> > > Another concern is the generalization of your defense to a variety of datasets and a variety of pre-trained LLMs.

---

> > > > ### Author Response · Authors · 2025-12-03
> > > > **Reply to Reviewer BprZ (Follow-up, Part 2)**
> > > >
> > > > > **W7 (more datasets and LLMs).** Another concern is the generalization of your defense to a variety of datasets and a variety of pre-trained LLMs.
> > > >
> > > > **A7.**
> > > > In our submission,
> > > > we **have already validated** the generalization of SPARD across two fine-tuning datasets (GSM8K and OpenbookQA) and two pretrained LLMs (Qwen-2.5-7B-Instruct and LLaMA-3.2-3B-Instruct).
> > > > To further address your concerns, we additionally evaluate SPARD on a new downstream task, **StrategyQA** [3], and on another LLM, **SmolLM2-1.7B-Instruct** [4].
> > > >
> > > > **(i) Generalization across tasks (StrategyQA).**
> > > > The table below shows the performance comparison on StrategyQA under harmful fine-tuning attacks.
> > > > As shown, **SPARD consistently achieves the lowest ASR across all attacks** while preserving comparable utility.
> > > > These results further demonstrate that SPARD **generalizes to diverse downstream tasks**, maintaining robustness across domains.
> > > >
> > > > | Method    | BeaverTails | I-BeaverTails | LatHarmful | Q-LatHarmful | **Avg. ASR ↓** || StrategyQA Acc ↑ |
> > > > | --------- | ----------- | ------------- | ---------- | ------------ | -------------- |-|------ |
> > > > |SFT| 76.60 | 77.57 | 86.06 | 89.41 | 82.41 || 72.05|
> > > > |PTST| 48.60 | 53.04 | 38.18 | 50.71 | 47.63 || **73.58** |
> > > > |SafeInstr | 68.00 | 71.07 | 74.95 | 75.97 | 72.50 || 70.63|
> > > > |Antidote|34.20 | 43.40 | 8.89 | 8.35 | 23.71 || 66.59|
> > > > |Constrained SFT |25.20 | 34.38 | 6.06 | **6.52** | 18.04 || 64.52|
> > > > |Lisa| 24.80 | 33.33 | 8.89 | 10.59 | 19.40 || 64.74|
> > > > | **SPARD** | **22.20** | **16.56** | **2.83** | 8.76 | **12.59** ||73.14|
> > > >
> > > > **(ii) Generalization across LLMs (SmolLM2).**
> > > > The table below shows the performance comparison on **SmolLM2-1.7B-Instruct** under harmful fine-tuning attacks.
> > > > Consistent with our main results, SPARD achieves the lowest ASR, outperforming other baselines by a large margin (e.g., reducing Avg. ASR from ~89% in Lisa to 9.67%).
> > > > Moreover, SPARD maintains GSM8K accuracy comparable to SFT, confirming that our SPAG optimization and DPP-based selection provide robust defense mechanisms **across different model architectures** without compromising utility.
> > > >
> > > > | Method    | BeaverTails | I-BeaverTails | LatHarmful | Q-LatHarmful | **Avg. ASR ↓** || GSM8K Acc ↑ |
> > > > | --------- | ----------- | ------------- | ---------- | ------------ | -------------- |-|------ |
> > > > |SFT| 85.20 | 84.91 | 96.77 | 95.11 | 90.50 || **50.30**|
> > > > |PTST| 76.00 | 80.50 | 89.09 | 94.30 | 84.97 || 49.75 |
> > > > |SafeInstr | 67.80 | 64.99 | 86.87 | 84.73 | 76.10||50.06|
> > > > |Antidote| 82.00 | 86.79 | 90.91 | 94.50 | 88.55 || 43.61 |
> > > > |Constrained SFT | 82.80 | 89.31 | 89.90 | 93.08 | 88.77| | 40.47 |
> > > > |Lisa| 83.40 | 89.73 | 91.92 | 91.24 | 89.07 || 42.29 |
> > > > | **SPARD** | **15.20** | **17.40** | **3.03** | **3.05** | **9.67** || 49.70|
> > > >
> > > >
> > > >
> > > > [3] *Did Aristotle Use a Laptop? A Question Answering Benchmark with Implicit Reasoning Strategies, TACL 2021.*
> > > >
> > > > [4] *SmolLM2: When Smol Goes Big -- Data-Centric Training of a Small Language Model, arXiv.*

---

> ### Comment · Reviewer_41kS · 2025-11-27
> **Gentle comments from Reviewer 41kS**
>
> Hi Reviewer BprZ ,
>
> I agree with the two concerns over the generalization on various datasets and pre-trained LLMs, and also lacking comparison with SOTA defense, which are very important to validate the effectiveness of the methods, and also apparantly this paper is not handling very well.
>
> I think a possible workaround is for the authors to implement these comparison in the rebuttal time-window (especially baseline defenses and experient on other architecture excluding LLama and Qwen) before end of the rebuttal. Do you think that would be a possible workaround?
>
> I provide two defenses that appear in ICML2025 and ICLR 2025 for reference.
>
> [1] Safety alignment should be made more than just a few tokens deep (ICLR2025)
>
> [2] Antidote: Post-fine-tuning safety alignment for large language models against harmful fine-tuning (ICML2025)

---

> ### Author Response · Authors · 2025-12-03
> **Reply to Reviewer BprZ (Follow-up, Part 1)**
>
> > **W5.** I only confirmed that I read your clarification, especially on promoting the strength of your approach.
>
> **A5.**
> We appreciate your follow-up and believe that our rebuttal has already addressed the concerns raised in your initial review. Your latest comment no longer revisits those points, but instead introduces **three new requests** (additional baselines, datasets, and backbones).
> These new requests are **generic and not specific to our method**, and could be raised to almost any submission regardless of technical soundness.
>
> To ensure a constructive discussion, we have nevertheless conducted and reported the requested additional experiments, although we believe our initial experimental results were already sufficient to validate our core contribution.
>
> ---
>
> > **W6 (more baselines).**  There are also recent work on harmful fine-tuning, including ICML 2025, ICLR 2025. Your comparison to only a very small number of defense methods. LISA appeared in 2024 and most methods in 2025 have shown better defense than LISA.
>
> **A6.**
> To address your additional concern regarding **more baselines**, we extend our evaluation to compare SPARD with two representative 2025 methods: (i) Antidote [1] (ICML 2025) and (ii) Constrained SFT [2] (ICLR 2025) on the GSM8K setting for Qwen-2.5-7B-Instruct.
>
> As shown in the table below, SPARD **consistently achieves the lowest ASR** across all attacks, while also obtaining **substantially higher fine-tuning accuracy** (>7%). This demonstrates that SPARD provides strong safety performance without sacrificing task utility, whereas Antidote and Constrained SFT trade off utility for safety.
>
> | Method    | BeaverTails | I-BeaverTails | LatHarmful | Q-LatHarmful | **Avg. ASR ↓** || GSM8K Acc ↑ |
> | --------- | ----------- | ------------- | ---------- | ------------ | -------------- |-|------ |
> |Antidote [1]|29.00 | 30.82 | 6.46 | 10.79 | 19.27 || 78.26 |
> |Constrained SFT [2]|23.80 | 35.43 | 7.47 | 24.00 | 22.68 || 78.10|
> | SPARD | **10.60**   | **14.05**     | **6.46**   | **10.39**    | **10.38**      || **85.77**   |
>
>
> [1] *Antidote: Post-fine-tuning safety alignment for large language models against harmful fine-tuning, ICML 2025*
>
> [2] *Safety alignment should be made more than just a few tokens deep, ICLR 2025*

---

### Official Review · Reviewer_Jg8Y · 2025-11-04

**Soundness:** 3
**Presentation:** 2
**Contribution:** 2
**Rating:** 4
**Confidence:** 2

**Summary:**

This paper proposes SPARD, a defense framework against harmful fine-tuning attacks on aligned large language models (LLMs). The framework combines two complementary ideas:
(1) Safety-Projected Alternating Gradient (SPAG) — an optimization procedure that alternates between utility-driven fine-tuning steps and explicit safety projection steps to enforce safety constraints in closed form;
(2) Relevance–Diversity Determinantal Point Process (DPP) — a principled method to select a compact subset of “safe” data that balances task relevance and behavioral diversity.

Experiments on GSM8K and OpenBookQA under multiple harmful fine-tuning settings (BeaverTails, LatHarmful, etc.) demonstrate that SPARD significantly reduces attack success rate (ASR) while maintaining downstream utility. The method consistently outperforms strong baselines such as SafeInstr, SafeLoRA, and Lisa.

**Strengths:**

1. The combination of task relevance and diversity within a determinantal point process framework is well-motivated. It improves upon prior ad-hoc or random selection strategies, and empirical analysis (Figure 2) convincingly shows that both factors matter for robust defense.
2. The mathematical formulation of SPAG (Eq. 2–4) and its derivation (Appendix A) are precise and well-presented. The trust-region stabilization strategy further enhances practical robustness.

**Weaknesses:**

- The proposed Safety-Projected Alternating Gradient (SPAG) method is presented as a novel optimization framework. However, its core mechanism is well-established in classical constrained optimization literature (e.g., projected gradient descent, proximal updates, or trust-region projection). While the adaptation to LLM safety alignment is interesting and practically meaningful, the methodological novelty of SPAG itself appears limited. The paper would benefit from a clearer distinction between conceptual novelty (application to harmful fine-tuning) and methodological novelty (new optimization formulation).
- Although the DPP-based selection is implemented with a greedy approximation, the paper does not provide a clear discussion or analysis of its computational efficiency.

**Questions:**

Please refer to the Weaknesses.

---

> ### Author Response · Authors · 2025-11-21
> **Reply to Reviewer Jg8Y (Part 1)**
>
> Thank you for the insightful review. We address your concerns as follows.
>
> > **W1.** The proposed Safety-Projected Alternating Gradient (SPAG) method is presented as a novel optimization framework. However, its core mechanism is well-established in classical constrained optimization literature (e.g., projected gradient descent, proximal updates, or trust-region projection). While the adaptation to LLM safety alignment is interesting and practically meaningful, the methodological novelty of SPAG itself appears limited. The paper would benefit from a clearer distinction between conceptual novelty (application to harmful fine-tuning) and methodological novelty (new optimization formulation).
>
> **A1.**
> We thank the reviewer for the insightful comment.
> We clarify that the contribution of SPAG does not lie in introducing a new generic projection operator.
> Rather, the novelty is in formulating safety-aligned fine-tuning as a **safety-constrained optimization problem** and in deriving a safe-specific **two-step procedure** directly from this formulation. We detail the novelty of SPAG as follows:
>
> **(i) Novel Safety-Constrained Formulation.**
> We are the first to cast LLM safety alignment as the constrained problem:
> $$\min\_{\mathbf{\theta}} \mathcal{L}(\mathcal D\_{\text{ft}}, \mathbf{\theta})\quad\text{s.t.}\quad \mathcal L(\mathcal D\_{\text{safe}}, \mathbf{\theta})\le \tau,$$
> Here, the constraint directly encodes a safety condition; it restricts the model to remain within a region where the safety loss does not exceed a specified threshold $\tau$. This yields a well-defined feasible set and a clear constraint boundary that are intrinsic to LLM safety.
>
> Previous works study safety by adding a penalty term as:
> $$\min\_{\mathbf{\theta}} \mathcal{L}(\mathcal D\_{\text{ft}}, \mathbf{\theta}) + \lambda\big(\mathcal{L}(\mathcal D\_{\text{safe}}, \mathbf{\theta}) - \tau\big)$$
> where a penalty parameter $\lambda \ge 0$ trades off utility and safety.
> Penalty methods only enforce safety **indirectly**, allowing safety constraint violations as long as they are offset in the objective and thus providing no explicit guarantee that the model stays within the safe region.
> In contrast, our formulation **enforces** the safety condition directly. By explicitly defining the feasible region through the safety loss, we obtain a principled structure that enables tractable projection-based optimization tailored specifically for maintaining LLM safety during fine-tuning.
>
> **(ii) Alternating SPAG Procedure Derived From the Constraint.**
> Based on this formulation, we derive a principled alternating algorithm including
> **(a)** a **utility update** on $\mathcal{D}\_{\text{ft}}$, followed by
> **(b)** an explicit **safety projection** onto a locally linearized feasible region defined by $\mathcal{L}\_{\text{safe}}$.
> The resulting update
> $\theta\_{\text{new}} = \theta^{+} - \frac{\mathcal{L}\_{\text{safe}}(\theta^{+}) - \tau} {|\nabla\mathcal{L}\_{\text{safe}}(\theta^{+})|^{2}} \nabla\mathcal{L}\_{\text{safe}}(\theta^{+})$
> is not a generic projection rule; its form, direction, and scaling arise specifically from the safety constraint.
>
> We have updated the paper to make explicit that:
> * SPAG novelty lies in the **problem formulation** and
> * the **derivation of a safety-specific projection step**,
>   rather than the generic projection mechanism itself.

---

> ### Author Response · Authors · 2025-11-21
> **Reply to Reviewer Jg8Y (Part 2)**
>
> > **W2.** Although the DPP-based selection is implemented with a greedy approximation, the paper does not provide a clear discussion or analysis of its computational efficiency.
>
> **A2.**
> Thank you for pointing out the need for a clearer discussion of the computational efficiency of the greedy DPP selection. We discuss theoretical analysis and empirical evidence as follows, and have added it in the revision.
>
> **(i) Theoretical Analysis**:
>
> Let $N = \lvert \mathcal D\_{\text{safe}} \rvert$ be the size of the safe pool and let a small $k = \lvert \mathcal C \rvert$ be the target number of selected samples.
> In our experiments, $N \approx 17{,}000$ and $k = 236$, so $k \ll N$, i.e., only a small fraction of the safe pool is selected.
>
> At greedy step $m$ (with $m-1$ items already selected), we maintain the Cholesky factor $\widehat{\mathbf L}\_{\mathcal C\_{m-1}} = \mathbf C \mathbf C^\top$, where $\mathbf C \in \mathbb{R}^{(m-1)\times (m-1)}$.
> To evaluate the gain factor for all remaining candidates $i \notin \mathcal C\_{m-1}$, we first extract the cross-kernel block
> $$ \mathbf V\_{m-1} = \widehat{\mathbf L}\_{\mathcal D\_{\text{safe}}\setminus \mathcal C\_{m-1}, \mathcal C\_{m-1}} \in \mathbb{R}^{(N-m+1)\times (m-1)},$$
> and then solve the triangular system
> $$ \mathbf C \mathbf W\_{m-1}^\top = \mathbf V\_{m-1}^\top, $$
> where each column of $\mathbf W\_{m-1}$ corresponds to the vector $\mathbf w\_i$ used in the gain $\widehat{\mathbf L}\_{ii} - \\|\mathbf w\_i\\|\_2^2$.
>
> Following [1], we avoid repeatedly solving triangular systems from scratch.
> Instead, for each candidate item $i$, we maintain its Cholesky coordinates and gain:
> $$ \mathbf{w}\_i \in \mathbb{R}^{m-1}, \qquad d\_i^2 = \widehat{\mathbf{L}}\_{ii} - \lVert \mathbf{w}\_i \rVert\_2^2, $$
> and update them *incrementally* when a new element $j$ is added to $\mathcal{C}\_{m-1}$.
> The update for each remaining candidate item $i$ is
> $$ e\_i = \frac{
>         \widehat{\mathbf{L}}\_{ij} -
>         \langle \mathbf{w}\_i,\, \mathbf{w}\_j \rangle
>       }{
>         d\_j
>       }, \qquad
>     \mathbf{w}\_i \leftarrow [\mathbf{w}\_i,\, e\_i], \qquad d\_i^2 \leftarrow d\_i^2 - e\_i^2, $$
> which requires only an inner product of length $m-1$.
> Thus, each candidate update costs $\mathcal{O}(m)$, and the entire gain update at step $m$ costs
> $$ \mathcal{O}\left((N - m + 1)\, m\right). $$
>
> Summing over all greedy steps $m = 1,\dots,k$ yields
> $$ \sum\_{m=1}^{k} \mathcal{O}\left((N-m+1)\, m\right)
>     = \mathcal{O}(N k^2)
>     \qquad (N \gg k). $$
> Thus, the final time complexity is $\mathcal{O}(N k^2)$, i.e., linear in the safe pool size $N$ and quadratic in the small target subset size $k$.
>
> **(ii) Empirical Efficiency:**
>
> In addition to theory, we benchmarked our implementation on a single NVIDIA A800 GPU, and the DPP selection for beaverTails attack takes only **0.12** seconds, which is negligible compared to LLM total training time **(≈4 hours)**.
>
> [1] Fast greedy map inference for determinantal point process to improve recommendation diversity, NeurIPS 2018.

---

### Comment · Area_Chair_nuWg · 2025-11-26
**Please Review Author Response**

Dear Reviewers,

The authors have now responded to your comments. Could you review their response as soon as possible? If you have any further questions or concerns, please raise them as well.

Best,

Your AC

---

### Author Response · Authors · 2025-12-03
**General Reply to ACs**

Dear ACs,

We sincerely thank you for your time in considering our paper and all reviewers for their thoughtful comments and feedback.
Below, we summarize the key strengths recognized by the reviewers, the main points discussed, and how our additional results addressed these concerns.

---
**SPARD** enforces safety constraints under harmful fine-tuning attacks by projecting updates into the safe subspace, complemented by a **relevance–diversity DPP** for selecting high-quality safety data.
Reviewers highlighted **three major strengths**:

**(i) Our Method is Well-motivated and Novel:** "well-motivated" (Jg8Y), "simple and interesting" (BprZ), "novel and principled solution" (pm8L), "the first (among a few concurrent work) to explore such projection methods" (41kS), "crucial and might inspire newer ideas on design defense" (41kS), "solid work" (41kS).

**(ii) Elegant Methodology:** "precise and well-presented" (Jg8Y), "elegant and makes perfect sense" (41kS), "enhances practical robustness" (Jg8Y), "extremely well written" (41kS).

**(iii) Strong Empirical Evidence.**
"extensive experiments, sensitivity analyses, and comparisons with baselines" (pm8L), "convincingly shows that both factors matter for robust defense" (Jg8Y), "consistently outperforms baseline methods across multiple datasets and model architectures." (pm8L), "nice and clear" (41kS).

---
### Discussion & our responses focus on the following points:

1. **Reviewer 41kS (Rating: 6 $\to$ 8, Confidence: 5)**
    - **Comparison with SafeGrad and CAGD:** We provide a detailed comparison, highlighting differences in triggering conditions and projection rules. Empirically, SPARD outperforms both methods.
    - **Comparison with "Your Task May Vary":** We clarify that this work considers only relevance, whereas SPARD integrates both relevance and diversity.
    - **Safety-Constraint Guarantee:** We empirically verify that the safety constraint remains satisfied throughout training.

    Reviewer 41kS **raised the score to 8**, stating that our rebuttal **"cleared up my concern"** and would **"be happy to champion this paper in review process"**.

2. **Reviewer pm8L (Rating: 6, Confidence: 3)**
    - **Efficiency analysis:** We clarify that the computational overhead is very small and include a complementary complexity analysis.
    - **Trigger Frequency:** We show that SPARD triggers safety backward only when necessary, thus maintaining training efficiency.
    - **Hyperparameter Sensitivity:** We show that SPARD is not sensitive to hyperparameters and provide practical recommendations.

    After the rebuttal, Reviewer pm8L **"didn't have any further question"** and maintained the positive score.

3. **Reviewer BprZ (Rating: 2, Confidence: 4)**

    **Concerns from the initial reviews:**
    - **Relevance–Diversity:** We explain the motivation for using relevance and diversity and how they jointly improve safety-data selection.
    - **Safe Samples Assumption:** We clarify that SPARD does not rely on a specific safe sample assumption, and all baseline methods use the same safe data.
    - **Boundary Cases:** We analyze the boundary cases and show that SPARD achieves a balanced and robust safety–utility trade-off.

    After reading our previous rebuttal, Reviewer BprZ stated that “Unfortunately, you have not addressed all concerns raised.” However, BprZ did **not** revisit the issues from the initial review; instead, they introduced three **entirely new, general requests**:
    - **Additional baseline, Dataset, LLM:** We extend our experiments to include two additional baselines (Antidote and Constrained SFT), one additional dataset (StrategyQA), and one additional LLM (SmolLM2). Across all settings, SPARD remains consistently the best-performing method.

4. **Reviewer Jg8Y (Rating: 4, Confidence: 2)**:
    - **Novelty of SPAG:** We clarify that SPAG is not a generic projection operator; it is novel in formulating the safety-constrained optimization problem and safe-specific two-step procedure.
    - **Efficiency analysis:** We provide detailed clarification showing that the runtime of SPAG is extremely small (within 1 second), along with an additional time-complexity analysis.


Best,

The Authors

---

### Meta-Review · Area_Chair_4Lrs · 2026-01-04

**Summary:**

This paper introduces SPARD, a defense mechanism against harmful fine-tuning attacks on Large Language Models (LLMs). The method is composed of two main parts: a safety-projected alternating gradient (SPAG) optimization strategy, which aims to steer model updates away from harmful directions, and a relevance-diversity data selection method using Determinantal Point Processes (DPP) to choose a small, effective subset of safety data. The authors conduct extensive experiments to show that SPARD can maintain model safety with minimal impact on task utility, claiming a superior trade-off compared to prior methods.

This paper presents a technically interesting approach to a significant problem and is supported by a large volume of experimental data. However, the review process has surfaced critical issues that make it difficult to recommend acceptance. The primary concern is the strong, persistent rejection from a highly confident expert reviewer (BprZ), whose concerns about novelty and the completeness of the evaluation, while demanding, are not without merit. While the authors' rebuttal was heroic in its effort, it ultimately failed to sway the most critical reviewer. Given this, the paper cannot be considered for publication in this ICLR.

**Reviewer Concerns:**

* *Novelty and Contribution:* The primary negative reviewer (BprZ) argued that the core technical components are extensions of existing paradigms (KL-divergence regularization, DPP) and that the overall contribution is therefore limited. This has not been fully addressed.

* *Generalization and Baselines:* A major point of contention was the scope of the evaluation. Reviewer BprZ, supported by Reviewer 41kS, raised concerns about the limited number of LLM architectures and datasets used, and the omission of comparisons to the most recent state-of-the-art defense methods. This has not been fully addressed.

**Reviewer Scores:**

* Reviewer BprZ: This reviewer would maintain their score of 2. Their position is firm.

* Reviewer 41kS and pm8L: Their lack of engagement is a critical unknown. While they initially scored the paper a 6, the intense and unresolved criticism from Reviewer BprZ could have potentially lowered their scores had they participated in a full discussion. It is not safe to assume their scores would have remained positive.

---

### Decision · Program_Chairs · 2026-01-26

Reject